# Identification of periodic attractors in Boolean networks using *a priori* information

Ulrike Münzner[1,2☯], Tomoya Mori[2☯], Marcus Krantz[3], Edda Klipp[3], Tatsuya Akutsu[2]*

**1** Institute for Protein Research, Laboratory of Cell Systems, Osaka University, Suita, Osaka, Japan, **2** Institute for Chemical Research, Bioinformatics Center, Kyoto University, Kyoto, Japan, **3** Institute of Biology, Theoretical Biophysics, Humboldt-Universität zu Berlin, Berlin, Germany

☯ These authors contributed equally to this work.
* takutsu@kuicr.kyoto-u.ac.jp

**Data Availability Statement:** The GitHub repository https://github.com/takutsu5/AttPrior contains the implemented algorithm and toy models. The original angiogenesis network is available at https://github.com/NathanWeinstein/

## Abstract

Boolean networks (BNs) have been developed to describe various biological processes, which requires analysis of attractors, the long-term stable states. While many methods have been proposed to detection and enumeration of attractors, there are no methods which have been demonstrated to be theoretically better than the naive method and be practically used for large biological BNs. Here, we present a novel method to calculate attractors based on *a priori* information, which works much and verifiably faster than the naive method. We apply the method to two BNs which differ in size, modeling formalism, and biological scope. Despite these differences, the method presented here provides a powerful tool for the analysis of both networks. First, our analysis of a BN studying the effect of the microenvironment during angiogenesis shows that the previously defined microenvironments inducing the specialized phalanx behavior in endothelial cells (ECs) additionally induce stalk behavior. We obtain this result from an extended network version which was previously not analyzed. Second, we were able to heuristically detect attractors in a cell cycle control network formalized as a bipartite Boolean model (bBM) with 3158 nodes. These attractors are directly interpretable in terms of genotype-to-phenotype relationships, allowing network validation equivalent to an *in silico* mutagenesis screen. Our approach contributes to the development of scalable analysis methods required for whole-cell modeling efforts.

## Author summary

Systems biology requires not only scalable formalization methods, but also means to analyze complex networks. Although Boolean networks (BNs) are a convenient way to formalize biological processes, their analysis suffers from the combinatorial complexity with increasing number of nodes $n$. Hence, the long standing $O(2^n)$ barrier for detection of periodic attractors in BNs has obstructed the development of large, biological BNs. We break this barrier by introducing a novel algorithm using *a priori* information. We

Angiogenesis-Model/blob/master/angiofull.net. The original cell cycle network is available as Supplementary Data 3 at https://doi.org/10.1038/s41467-019-08903-w. All remaining relevant data are within the manuscript and its Supporting information files.

**Funding:** This work was supported by the Japan Society for the Promotion of Science (JSPS) with a JSPS International Research Fellowship to UM (ID: PE17765). TA was partially supported by JSPS Grants-in-Aid for Scientific Research (KAKENHI) (Grant number 18H04413). The funders had no role in study design, data collection and analysis, decision to publish, or preparation of the manuscript.

**Competing interests:** The authors have declared that no competing interests exist.

show that the proposed algorithm enables systematic analysis of BNs formulated as bipartite models in the form of *in silico* mutagenesis screens.

## Introduction

Boolean network (BN) analysis is a powerful tool to computationally study biological processes, which include gene regulatory networks [1–3] and neural networks [4]. Given an initial state where each node (*e.g.*, each gene) is assigned a value zero (equivalent terms: 0, inactive, false) or one (equivalent terms: 1, active, true), the node values are updated according to the Boolean functions to compute the network state in the next time step. In a BN, two types of stable states are observed: *point attractors* (statically stable states) and *periodic attractors* (periodically stable states). Describing biological processes as BNs allows a qualitative, dynamic description, where the attractors can be interpreted as stable states of a cell [5, 6]. Furthermore, driving biological systems to stable states is also important and has been studied using both Boolean models [7] and neural network models [8]. However, which attractor is reached often depends on the initial state and thus, testing of all possible initial states may require an enormous computational burden because there exist $2^n$ states in a BN with $n$ nodes.

One approach to circumvent this problem is network reduction as in [5, 9]. Although the conservation of attractors has been shown for some methods [10, 11], this might not be the generic case. Existing approaches to calculate attractors in larger networks include [12, 13], and have been applied to large, random BNs. Attractor analyses of biological BNs have so far been limited to small networks, which might reflect the lack of feasible analysis methods. From a theoretical viewpoint, the attractor detection problem for BNs is NP-hard in general [3] and most existing algorithms do not have guaranteed time complexity bound less than $O(2^n)$. Although there exist some polynomial time algorithms for finding a point attractor, the target classes of BNs are very restricted ones [3, 14, 15]. For BNs consisting of AND/OR functions, $O(1.587^n)$ time and $O(1.985^n)$ time algorithms have been developed for finding a point attractor [16], and a periodic attractor with period 2 [17], respectively. These algorithms are based on cutting of unnecessary partial states. For example, suppose that node $y$ is activated only if node $x$ is activated. Then, we need not examine states including $(x, y) = (0, 1)$. However, for example, if an XOR function is assigned to $y$, this strategy does not work. The divide and conquer approach was also employed in [17], which shows that detection of a periodic attractor of a constant period can be done in polynomial time if the maximum degree is bounded by a constant and a given network is decomposable into small pieces (precisely, the treewidth of a given network is bounded by a constant). However, in some cases, BNs may not be decomposed into small pieces. For example, if a given BN is a clique (all nodes are connected by edges), there is no way to decompose it. Accordingly, to the best of our knowledge, there does not exist an algorithm with less than $O(2^n)$ time worst case or expected time complexity for finding a periodic attractor with period 3 or more for a reasonably wide class of BNs.

It is worthy to mention that many practically efficient methods have been developed for detection and/or enumeration of attractors using such techniques as logic programming [18], SAT solvers [12], binary decision diagrams [19], and answer set programming [20], as well as representation/approximation of complex attractors through stable motifs [21] (equivalently, symbolic steady states [22] and trap spaces [13]). Although these methods are practically very useful, there is no theoretical guarantee better than $O(2^n)$ on either worst-case or expected time complexity. It is also worthy to mention that extensive studies have been done on related problems on BNs [7, 23–25] using an algebraic approach, called semi-tensor product [2].

However, almost all methods used in these studies use $2^n \times 2^n$ or larger size matrices and no method has a guaranteed worst case or expected time complexity less than $O(2^n)$. Therefore, $O(2^n)$ is the long standing barrier. Furthermore, it is suggested in [3] that the attractor detection problem is PSPACE-hard if there is no limit on the size of attractors. This suggests further that detection of long attractors is harder than the class NP (unless NP = PSPACE) and thus practical solvers for NP-hard problems (*e.g.*, solvers for integer linear programming (ILP) and the Boolean satisfiability problem (SAT)) may not be effectively applied. Therefore, development of efficient algorithms for the detection of long attractors remains a great challenge.

In order to cope with the $O(2^n)$ time barrier, we propose in this study a novel approach, which makes use of *a priori* information. Suppose that we know that each bit in one global state in the target (point or periodic) attractor takes some specific value (0 or 1) with probability $p$, the *a priori* information. The *a priori* information reflects the expected state (0 or 1) of each node and the confidence in this state (a probability $0 \leq p \leq 1$). The first information can be inferred from binarization of experimental data, *e.g.* by assigning the value 1 to an expressed gene. We refer to this information as initial guess. The second part, the probability, is much more difficult to infer from experimental data. However, the beauty of the method is that this value can be tried out empirically: The higher the confidence, the later the value will be switched by the algorithm. Of course, there are also situations when the value can be inferred from data: *E.g.*, the gene can be detected (over a certain threshold) in 7 out of 10 time points: $p_{true} = 0.70$.

Then, we will show that the target attractor can be found within an $O([p^{1-\alpha}(1 - p)^\alpha \beta^2]^n \cdot n^2)$ expected number of trials, where $\alpha = 1/(1 + \sqrt{p/(1 - p)})$, $\beta = \left(\frac{1}{\alpha}\right)^\alpha \cdot \left(\frac{1}{1-\alpha}\right)^{1-\alpha}$. Under a reasonable assumption, the expected time complexity will be $O([p^{1-\alpha}(1 - p)^\alpha \beta^2]^n poly(n))$, where $poly(n)$ means some polynomial on $n$. This expected computational complexity is less than $O(2^n)$ for $p > 0.5$. For example, it is $O(1.600^n poly(n))$ for $p = 0.9$ and $O(1.917^n poly(n))$ for $p = 0.7$. The exponential factors on these complexities will be much smaller than $2^n$ as $n$ grows. Therefore, this result is an important theoretical contribution. Here, we briefly compare $2^n$ and $1.917^n n^2$. The ratio $2^n/(1.917^n n^2)$ is 0.00693 for $n = 100$, 0.1201 for $n = 200$, 6397.4 for $n = 500$, and $2.558 \times 10^{12}$ for $n = 1000$. Since it is expected that the polynomial factor is not so large, the proposed algorithm is much faster than the naive one if $n \geq 500$. It is worthy to note that if it is enough to find some attractor, there exists a trivial approach: start from a random initial state and then follow the trajectory until reaching an attractor. This method works efficient unless the trajectory is very long. However, it cannot control the type of a detected attractor, even for singleton or periodic. Therefore, *a priori* information would be useful to control the search towards detection of the desired attractor. It should also be noted that our algorithm examines $2^n$ initial states in the worst-case (i.e., *a priori* information is not at all useful) and thus does not improve the worst-case time complexity.

We utilize this novel algorithm for the analysis of two BNs which differ in size, modeling formalism, and biological scope.

## Angiogenesis network

First, we study a BN formulated in the classical way describing the effect of several microenvironments on the behavior of endothelial cells (ECs) during angiogenesis [9]. During angiogenesis, new blood vessels are formed by sprouting from existing ones, a process occurring during embryonic development but also tumor growth. Angiogenesis requires differentiation of ECs, constituting the innermost layer in a blood vessel, into one of three specialized behaviors: phalanx, stalk, and tip. Angiogenesis and EC specialization may be triggered by a hypoxic microenvironment [26]. In their study [9], the authors propose a BN which accounts for the

regulatory network triggering angiogenesis by inducing specialized EC behavior depending on the microenvironment. They first propose a full BN with $n = 142$ nodes, and reduce it to $n = 64$ nodes. The authors use the reduced model to identify attractors applying an adaptation of a SAT-based approach [12] provided in the BoolNet package [27], interpret which of these found attractors correspond to which EC behavior, and trace the microenvironment that triggered each of these behaviors. The authors define in total 16 nodes of the network which comprise the microenvironment. EC behavior is interpreted by the activity configuration of the four marker nodes *AKT1*, autocrine *JAG1*, *DLL4a*, and *NRP1*. A microenvironment induces typical behavior if all detected attractors show an EC marker configuration corresponding to a single and stable EC signature. Otherwise, the microenvironment is regarded as inducing atypical behavior. For example, if the EC marker signature does not correspond to any of the three behaviors or oscillates between them, the induced behavior is regarded as atypical. In the end, the authors find in total 35 microenvironments (configurations or patterns of activity of the 16 microenvironment nodes) which trigger typical EC behavior (phalanx, stalk, or tip behavior). They also find 32 microenvironments which induce atypical behavior, where microenvironments inducing an invalid EC marker configuration, or unstable signatures are regarded as atypical. For example, if a microenvironment results in a periodic attractor in which the EC markers are unstable (do not have a fixed value) the associated behavior is regarded as atypical. Details regarding EC behavior markers and the molecular characteristics of the microenvironments can be obtained from the original study [9], and in the Materials and methods section.

The authors deemed the original network with $n = 142$ nodes too large for attractor analysis, hence, the attractors of the full network remain unknown. This makes the full network an excellent target to demonstrate that our proposed algorithm can easily be applied to a network of this size. Furthermore, our analysis enables us to compare the results from two networks which have been proposed to contain the same information.

## Cell cycle control network

Second, we study the behavior of a cell cycle control network [28] formulated as a bipartite Boolean model (bBM). A bBM consists of two distinct types of nodes: State nodes and reaction nodes. The reaction nodes define the regulatory layer of the system, by defining which reactions can fire in the current regulatory state—given that their source states are or become available. Each state node captures a possible site-specific state of the components, such as (the absence of) a specific covalent modification, or a bond between two specific domains. The neutral complement of those states are the unmodified residues or unbound domains. That means that a network component may be described by several sets of state nodes, each corresponding to a specific residue or domain. At least one of those need to be true for a component to be present (*i.e.*, if the protein is present neither bound or unbound at specific domain, then the protein is absent). This will be important below.

The cell cycle network we analyze here comprises 3158 nodes and describes in mechanistic detail the pathways controlling cell division in *Saccharomyces cerevisiae*. This is a hybrid model, composed of two different layers: A signaling layer, and a layer corresponding to three macroscopic processes, spindle pole body duplication, DNA replication, and budding. The signaling layer is modeled in mechanistic detail, and controls the progression of the macroscopic processes, the progression of these, in turn, feeds back into the signalling layer. The macroscopic processes each follow a sequence of irreversible and mutually exclusive events which occur during one round of cell division cycle. These events are verbal descriptions of different cell states, such as *DNA licensed*, *DNA replicating*, etc. Per definition, one and only one of these must be true at any given time, as, *e.g.*, the DNA cannot be both replicating and

replicated at the same time. It is also important to note that most components are not being turned over in the model, *i.e.* the model does not include synthesis of new components, but only changes their states. This will also be important for the analysis below.

In the original study, the authors did not perform an attractor search. Instead, they started from a trivial initial condition where all components were true and present in their neutral (unmodified/unbound) states. From this highly artificial initial state, the authors simulated the model in the absence of *Nutrients* (one of the requirements for the $G_1$/S transition and a model input) until the model found its natural initial state—corresponding to G0 arrest. This indeed lead to a biologically relevant point attractor, that could be released into a biologically relevant periodic attractor by setting *Nutrients* to 1 (active), but the authors did not scan for other possible and possibly relevant attractors. This previously detected periodic attractor corresponds to the wildtype. From this wildtype attractor components were systematically and manually removed to represent a set of selected mutant genotypes. However, it remains unknown to what extent other attractors exist. To the knowledge of the authors, a biological bBM of the size of the cell cycle control network has not yet been analyzed using a semi-automatic approach to study the attractor landscape. Hence, the cell cycle control network is a good example to demonstrate the feasibility of our proposed algorithm to the analysis of large-scale BNs, as well as to explore if attractors others than the known ones exist.

We study the robustness of these two networks by perturbing them with functional mutations. In addition, we explore attractor dynamics of the cell cycle network upon knockout mutations and combinations with functional mutations.

## Boolean network

We briefly introduce the formal concept of BNs. A BN consists of a set $V = \{x_1, \ldots, x_n\}$ of nodes and a list $F = (f_1, \ldots, f_n)$ of *Boolean functions*. For each node $x_i$, $f_i$ is assigned along with a set of inputs nodes $IN(x_i) = \{x_{i_1}, \ldots, x_{i_{k_i}}\}$, where $k_i$ is called the *indegree*. The state of node $x_i$ at time $t$ is denoted by $x_i(t)(\in \{0, 1\})$. Accordingly, the state of the whole BN at time step $t$ is represented by a binary vector $\mathbf{x}(t) = [x_1(t), \ldots, x_n(t)]$. The state of node $x_i$ at time $t + 1$ is determined by $x_i(t + 1) = f_i(x_{i_1}(t), \ldots, x_{i_{k_i}}(t))$. Here, we consider a synchronous BN (*i.e.*, the states of nodes are updated simultaneously) and thus the dynamics of a BN can be represented as $\mathbf{x}(t + 1) = \mathbf{f}(\mathbf{x}(t))$. The network structure of a BN is represented by a directed graph $G(V, E)$ such that $E = \{(x_{i_j}, x_i)|x_{i_j} \in IN(x_i)\}$. The dynamics of a BN can be well represented by a *state transition diagram*, in which a node and a directed edge correspond to a state of the BN and a state transition, respectively. Starting from any initial state, a BN will eventually reach a periodic sequence of global states $\langle \mathbf{x}(t), \mathbf{x}(t + 1), \cdots, \mathbf{x}(t + t_p - 1)\rangle$, where $t_p \geq 1$ and $\mathbf{x}(t + t_p) = \mathbf{x}(t)$. This sequence is called an *attractor*, and $i_p$ is called the *period*. An attractor is called a *point attractor* if $t_p = 1$, and a *periodic attractor* otherwise.

For example, consider a BN defined by

$$\begin{aligned}
x_1(t + 1) &= x_3(t), \\
x_2(t + 1) &= x_1(t) \wedge \overline{x_3(t)}, \\
x_3(t + 1) &= x_1(t) \wedge \overline{x_2(t)},
\end{aligned}$$

where $x \wedge y$ and $\overline{x}$ denote the conjunction (AND) of $x$ and $y$ and the negation (NOT) of $x$, respectively. Then the structure of a BN and its state transition diagrams are illustrated in Fig 1. This BN contains two point attractors and one periodic attractor with period 2. This type of BN refers to the classical approach, where a gene is represented by a single node. This means

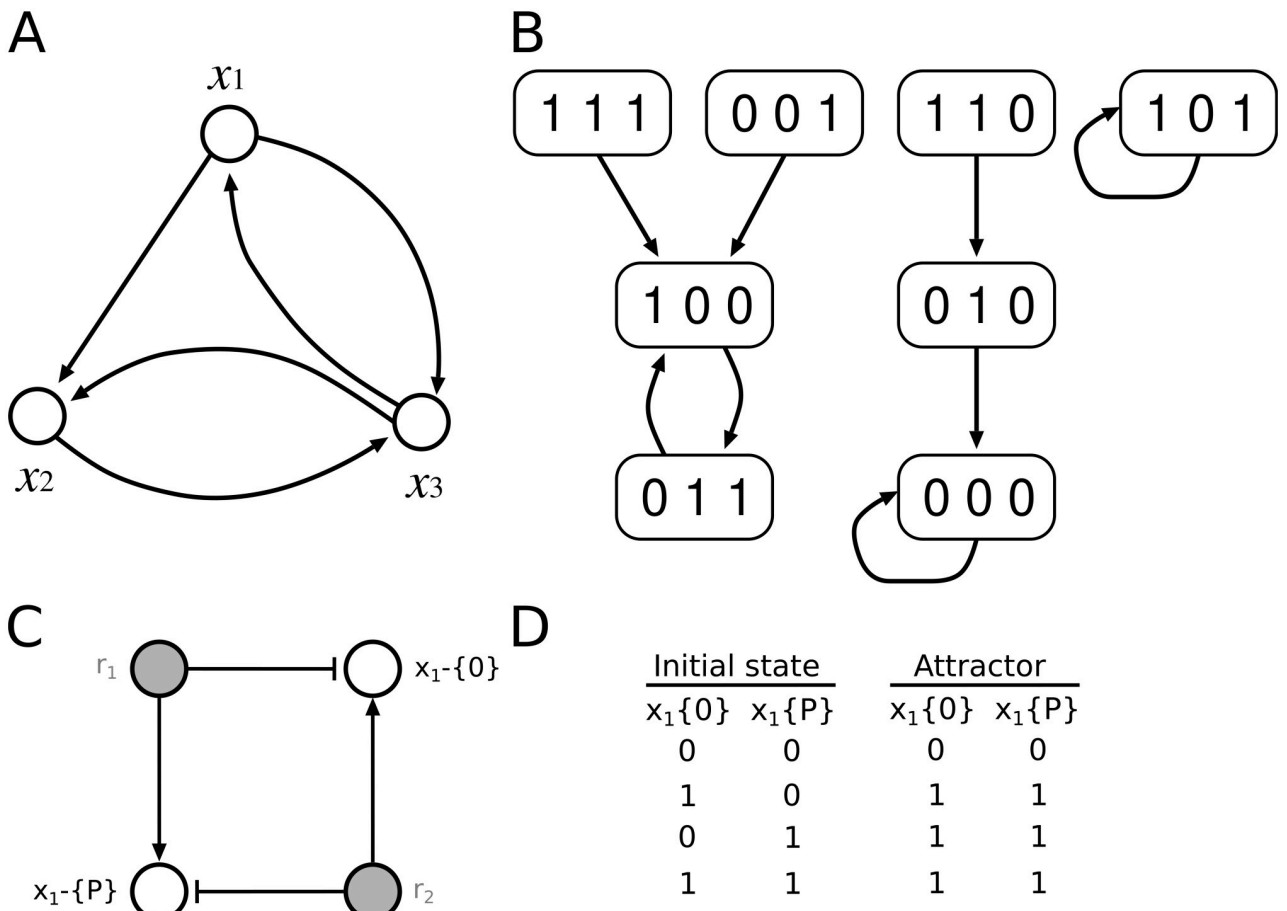

**Fig 1. Example of a BN.** (A) Network structure $G(V, E)$ for a BN consisting of $n = 3$ nodes $x_1$, $x_2$, and $x_3$. The Boolean functions are represented by the edges. (B) State transition diagram of all $2^3 = 8$ states of the BN represented in (A). (C) Simplification of a bBM for a phosphorylation-dephosphorylation system as elemental species-reaction graph. The nodes corresponding to elemental states of component $x_1$ are shown in black, the nodes corresponding to reactions are shown with grey fillings. Here, a component (protein) $x_1$ undergoes phosphorylation by consuming its elemental state $x_1$-{0} to produce its modified state $x_1$-{P} by reaction $r_1$ (phosphorylation). The modified $x_1$-{P} undergoes consumption thereby producing the neutral state $x_1$-{0} via reaction $r_2$ (dephosphorylation) (details in [29]). (D) Relationship between initial states of both elemental states $x_1$-{0} and $x_1$-{P} of component $x_1$ and the attractors. When at least either of the two elemental states $x_1$-{0} or $x_1$-{P} are active in the initial state, both will be active in the attractor. Both elemental states need to be inactive in the initial state to be absent (inactive) in the attractor state. Both reaction nodes $r_1$ and $r_2$ are assumed to be active in this example and are not shown. Synthesis and degradation are not considered here. In the example, standard update rules are applied, where two mutually exclusive states can be simultaneously present. More details can be found in [29].

that a single node representing a gene implicitly represents all biochemical modifications the gene can undergo.

A special form of BNs are bipartite Boolean models (bBMs, simplified example in Fig 1C). While they, like classical BNs, consist of nodes and vertices, the modifications of a single gene can be expressed explicitly and in mechanistic detail. The bipartite network structure has two types of nodes, state nodes, corresponding to elemental states (*i.e.* empirical observables such as specific phosphorylations), and reaction nodes, corresponding to elemental reactions (*i.e.* decontextualized biochemical reaction events). State nodes influence reaction nodes, and hence, all incoming edges of a reaction node describe how a reaction is regulated, allowing a mechanistic description of biological processes (details in [29, 30]).

The two types of Boolean modeling formalism, classical and bipartite, have different consequences for the perturbation analysis. In a classical BN, a gene or component can be knocked

out by disabling the node it corresponds to. In a bBM, most components are represented by multiple nodes corresponding to different states at one or more distinct sites (residues or domains). Also, some, but not all, may be represented as genes, including transcription and translation reactions. However, for many components, the model used here only includes posttranslational modifications and complex formation. At each level, there may be one or more elemental states involved which are all represented by individual nodes. Hence, there is a one-to-many relationship between a gene and the number of nodes which represent the different elemental states of the gene in a bBM. Conversely, if all states corresponding to a single residue or domain are set to false, then the component has *de facto* been deleted as long as no synthesis occurs in the model. This has specifically implications for the interpretation of detected attractors in the cell cycle control network. Our proposed algorithm perturbs the node values based on probabilities, and hence sometimes creates deletion mutants by removing the gene or by removing the only remaining state for a specific site or domain in a protein. In both cases, the modification has effectively altered the genotype of the model. Similarly, an essential component such as RNA polymerase II could be removed, or one of the macroscopic states could be added or removed, leading to a model that is no longer biologically meaningful. The latter is rare, and we consider it a technical artifact. This also explains why a bBM such as the cell cycle control network cannot be simulated with asynchronous updates. Since components are represented by multiple states that get produced or consumed, a reaction may consume (remove) a source state and produce (add) a product state. If the two states are updated independently, a consumption update that is not matched by a production update would eventually result in the removal of the only remaining state of a specific site or domain, and hence, resulting in the effective deletion of that component. The synchronous updates ensure that this cannot happen.

## Results

### Algorithm

We begin with simple examples to explain the basic idea of our proposed algorithm using *a priori* information.

First, we consider the simplest case of $n = 1$ and the target attractor is a point attractor such that $x_1 = b_1$ (we omit time step $t$ because we consider a point attractor), where $b_1$ corresponds to a point attractor and takes 0 (resp., 1) with probability 0.5. Suppose that we examine $x_1 = 1$ first, and then $x_1 = 0$. Since the first trial succeeds with probability 0.5, the number of expected trials is $\frac{1}{2} \cdot 1 + \frac{1}{2} \cdot 2 = \frac{3}{2}$. On the other hand, suppose that we know that $b_1 = 0$ holds with probability $\frac{3}{4}$. In this case, we examine $x_1 = 0$ first, and then $x_1 = 1$. Then, the expected number of trials is $\frac{3}{4} \cdot 1 + \frac{1}{4} \cdot 2 = \frac{5}{4}$, which is less than $\frac{3}{2}$.

Next, we extend this example for the case of $n = 2$. Suppose that the target attractor is a point attractor such that $x_i = b_i$ ($i = 1, 2$), where each $b_i$ takes 0 (resp., 1) with probability 0.5. If there is no prior information, we examine all 2-bit patterns in any order, for example, 11,10,01,00. Then, the expected number of trials is $\frac{1}{4}(1 + 2 + 3 + 4) = \frac{5}{2}$. On the other hand, suppose that we know that $b_i = 0$ holds with probability $\frac{3}{4}$ for each $i \in \{1, 2\}$. In this case, we examine 2-bit patterns in the following order: 00,01,10,11. Then, the expected number of trials is $\left(\frac{3}{4}\right)^2 + \left(\frac{3}{4}\right) \cdot \left(\frac{1}{4}\right) \cdot 2 + \left(\frac{1}{4}\right) \cdot \left(\frac{3}{4}\right) \cdot 3 + \left(\frac{1}{4}\right)^2 \cdot 4 = \frac{7}{4}$, which is smaller than $\frac{5}{2}$.

We can extend this idea to general $n$. The algorithm is quite simple (although its analysis is involved). It starts testing for attractors from the most plausible vector $\mathbf{x}_0 \in \{0,1\}^n$ with following the trajectory starting from it. If the search from this vector fails, the algorithm examines trajectories from the vectors each of which has one bit different from the first one. It further

repeats the same procedure by increasing the number of bits different from the first vector. The following gives the formal description of the algorithm, where we assume without loss of generality (w.l.o.g.) that the prior probability of 0 is larger than that of 1 for all bits (otherwise, it is enough to exchange the roles of 0 and 1 for the corresponding bits).

1. For $k = 0$ to $n$, do STEP 2 and 3.

2. For all $\mathbf{x}_0 \in \{0,1\}^n$ that contains $k$ bits with value 1, do STEP 3.

3. Let $\mathbf{x}(0) = \mathbf{x}_0$. Compute $\mathbf{x}(t + 1) = \mathbf{f}(\mathbf{x}(t))$ repeatedly until $\mathbf{x}(t + 1) = \mathbf{x}(t')$ holds for some $t' \leq t$ (which means $\langle \mathbf{x}(t'),\ldots, \mathbf{x}(t)\rangle$ is an attractor). If the attractor is a desired one, output it and exit.

In the following, we refer to this algorithm as **ATTapriori**. Note that how to decide whether the current attractor is a desired one is not a trivial task. If we know some exact criteria (e.g., allowable range of the attractor period and/or states of specific nodes), we can add a subroutine to check it. Otherwise, the algorithm can be terminated if the number of trials exceeds a specified number or CPU time exceeds the time limit. Then, we may manually check (maybe with some user-customized computer program to rule out obviously non-desired ones) whether or not there exists a desired one among the attractors reported so far.

It is to be noted that the algorithm can be modified so that it can enumerate all attractors by removing "If the attractor is a desired one, output it and exit." of STEP 3 and merging the identical attractors. However, it would take more than $O(2^n)$ time (because it will examine all $2^n$ starting vectors) and thus the resulting algorithm is meaningless. Since the purpose of this algorithm is to make use of *a priori* information on some global state in a specific attractor, it is reasonable that the algorithm is not useful for enumerating all attractors.

## Time complexity analysis

Here we analyze the expected time complexity of **ATTapriori**.

We consider a BN with $n$ nodes. Suppose that $x_i = 0$ holds with probability $p$ for all $i = 1, \ldots, n$ in some specific global state $\mathbf{x}_t$ of the target attractor, where $p > 0.5$. Note that if $x_i = 1$ has the probability $p$ ($p > 0.5$) it is enough to exchange the roles of 0 and 1 for such nodes. In theoretical analyses, the objective is to give an upper bound of the number of trials until the algorithm examines $\mathbf{x}_t$ as a starting vector. Therefore, we need to modify STEP 3 as follows:

If $\mathbf{x}_0 = \mathbf{x}_t$, output it and exit.

Of course, this modified algorithm is meaningless in practice because it is impossible to know $\mathbf{x}_t$ in advance. However, the expected number of trials (i.e., the expected number of tested $\mathbf{x}_0$s) for this modified algorithm gives an upper bound of that for the original algorithm because it examines the whole trajectory beginning from $\mathbf{x}_0$ (much more than one global state $\mathbf{x}_0$ per trial).

**ATTapriori** examines bit vectors from those with a smaller number of 1s to a larger number of 1s (e,g., 000, 001, 010, 100, 011, 101, 110, 111) where ties can be broken in an arbitrary manner. We can see that the expected number of trials $E(n, p)$ until reaching a desired attractor is given by

$$E(n,p) \;=\; \sum_{k=0}^{n} \left[ p^{n-k} \cdot (1 - p)^k \cdot \sum_{j=1}^{\binom{n}{k}} (S(n, k) + j) \right],$$

where

$$S(n, k) \quad = \quad \sum_{i=0}^{k-1} \binom{n}{i}.$$

Note that $p^{n-k} \cdot (1-p)^k$ is the success probability per vector $\mathbf{x}_0$ containing $k$ bits with value 1, and $S(n, k) + j$ denotes the number of trials (i.e., the number of different staring vectors) until the current vector (i.e., $S(n, k) + j$th vector) is examined.

**Lemma 1**. *The expected number of trials $E(n, p)$ is $O(f(\alpha^*)^n \cdot n^2)$, where $\alpha^* = \dfrac{1}{1 + \sqrt{\frac{p}{1-p}}}$,*

$f(\alpha) = p^{1-\alpha}(1-p)^\alpha \beta^2$, *and* $\beta = \left(\dfrac{1}{\alpha}\right)^\alpha \cdot \left(\dfrac{1}{1-\alpha}\right)^{1-\alpha}$.

*Proof.* In order to analyze the order of $E(n, p)$, we divide this expectation due to vectors $\mathbf{x}_0$ containing at most $\lceil n/2 \rceil$ bits with value 1 (i.e., at most half of the bits have value 1) and those containing at least $\lceil n/2 \rceil + 1$ bits with value 1. First, we evaluate the former part, that is, the partial sum $F(n, p)$ of $E(n, p)$ defined by

$$F(n, p) \quad = \quad \sum_{k=0}^{\lceil n/2 \rceil} \left[ p^{n-k} \cdot (1-p)^k \cdot \sum_{j=1}^{\binom{n}{k}} (S(n, k) + j) \right].$$

Since $\binom{n}{k-1} < \binom{n}{k}$ holds for $k \leq \lceil n/2 \rceil$, we have

$$\sum_{j=1}^{\binom{n}{k}} (S(n, k) + j) \quad < \quad \sum_{j=1}^{\binom{n}{k}} \left[ n \cdot \binom{n}{k} \right]$$
$$= \quad n \cdot \binom{n}{k}^2.$$

Thus, we have

$$F(n, p) \quad < \quad \sum_{k=0}^{\lceil n/2 \rceil} \left[ p^{n-k} \cdot (1-p)^k \cdot n \cdot \binom{n}{k}^2 \right]$$
$$< \quad n^2 \cdot \max_{k=1,\dots,\lceil n/2 \rceil} \left\{ p^{n-k} \cdot (1-p)^k \cdot \binom{n}{k}^2 \right\}$$

Next, we need to find an upper bound of 'max' in the last part of the above inequality. To this end, we let $k = \alpha n$. By using upper and lower bounds of Stirling's approximation

$\sqrt{2\pi n}\left(\frac{n}{e}\right)^n \le n! \le e\sqrt{n}\left(\frac{n}{e}\right)^n$, we have

$$
\binom{n}{\alpha n} \le \frac{e\sqrt{n}\left(\frac{n}{e}\right)^n}{\left(\sqrt{2\pi \alpha n}\left(\frac{\alpha n}{e}\right)^{\alpha n}\right) \cdot \left(\sqrt{2\pi(1-\alpha)n}\left(\frac{(1-\alpha)n}{e}\right)^{(1-\alpha)n}\right)}
$$

$$
= c \cdot \frac{\left(\frac{n}{e}\right)^n}{\left(\frac{\alpha n}{e}\right)^{\alpha n} \cdot \left(\frac{(1-\alpha)n}{e}\right)^{(1-\alpha)n}}
$$

$$
= c \cdot \frac{n^n}{(\alpha n)^{\alpha n} \cdot ((1-\alpha)n)^{(1-\alpha)n}}
$$

$$
= c \cdot \left[\left(\frac{1}{\alpha}\right)^\alpha \cdot \left(\frac{1}{1-\alpha}\right)^{(1-\alpha)}\right]^n,
$$

where $c = \frac{e}{\sqrt{4\pi^2\alpha(1-\alpha)n}}$. Here we define $\beta$ by

$$
\beta \equiv \left(\frac{1}{\alpha}\right)^\alpha \cdot \left(\frac{1}{1-\alpha}\right)^{1-\alpha}.
$$

Then, the exponential factor of $F(n, p)$ is bounded by $\max_{\alpha \le 0.5} f(\alpha)^n$ where

$$
f(\alpha) = p^{1-\alpha}(1-p)^\alpha \beta^2.
$$

Since it is difficult to directly maximize $f(\alpha)$, we derive $\alpha$ maximizing $g(\alpha)$ where

$$
\begin{aligned}
g(\alpha) &\equiv \ln f(\alpha) \\
&= (1-\alpha)\ln p + \alpha \ln(1-p) \\
&\quad - 2\alpha \ln \alpha - 2(1-\alpha)\ln(1-\alpha).
\end{aligned}
$$

By differentiating $g(\alpha)$ with respect to $\alpha$, we have

$$
g\prime(\alpha) = -\ln p + \ln(1-p) - 2\ln \alpha + 2\ln(1-\alpha).
$$

By letting $g'(\alpha) = 0$, we have

$$
\begin{aligned}
\ln\left(\frac{1-p}{p}\right) - \ln\left(\frac{\alpha}{1-\alpha}\right)^2 &= 0, \\
\left(\frac{1-p}{p}\right) &= \left(\frac{\alpha}{1-\alpha}\right)^2, \\
\sqrt{\left(\frac{1-p}{p}\right)} &= \frac{\alpha}{1-\alpha}.
\end{aligned}
$$

By solving the last equality, we have

$$
\alpha* = \frac{1}{1 + \sqrt{\dfrac{p}{1-p}}}.
$$

This $\alpha^*$ means that

$$\max_{k=1,\ldots,\lceil n/2 \rceil}\left\{ p^{n-k} \cdot (1-p)^k \cdot \binom{n}{k}^2 \right\} \;\leq\; c \cdot \left[ p^{1-\alpha^*}(1-p)^{\alpha^*}\beta^2 \right]^n$$

$$= \; c \cdot \left[ p^{1-\alpha^*}(1-p)^{\alpha^*}\left(\frac{1}{\alpha^*}\right)^{2\alpha*} \right]^n.$$

Therefore, $F(n,p)$ is

$$O\!\left( \left[ p^{1-\alpha^*}(1-p)^{\alpha^*}\left(\frac{1}{\alpha^*}\right)^{2\alpha^*}\left(\frac{1}{1-\alpha^*}\right)^{2(1-\alpha^*)} \right]^n \cdot n^2 \right).$$

Furthermore, we can verify that $\alpha^* < \frac{1}{2}$ holds. In order to evaluate the partial sum of $E(n,p)$ due to vectors containing at least $\lceil n/2 \rceil + 1$ bits with value 1,

$$S(n,\lceil n/2 \rceil) \;>\; \gamma 2^n$$

holds for some constant $\gamma > 0$. Since $S(n,k) \leq 2^n$ holds for any $k \leq n$,

$$\sum_{j=1}^{\binom{n}{k}} (S(n,k)+j) \;\leq\; 2 \cdot 2^{2n} \;\leq\; d \sum_{j=1}^{\binom{n}{\lceil n/2 \rceil}} (S(n,k)+j)$$

holds for some constant $d$. Thus, by letting $c_{n,k} = \binom{n}{k}$, the partial sum of $E(n,p)$ due to the latter half of vectors is bounded as

$$\sum_{k=\lceil n/2 \rceil+1}^{n} \left[ p^{n-k} \cdot (1-p)^k \cdot \sum_{j=1}^{c_{n,k}} (S(n,k)+j) \right]$$

$$\leq \; d \cdot \left[ \sum_{j=1}^{c_{n,\lceil n/2 \rceil}} (S(n,k)+j) \right] \cdot \sum_{k=\lceil n/2 \rceil+1}^{n} \left[ p^{n-k} \cdot (1-p)^k \right]$$

Therefore, the expected number of trials $E(n, p)$ is bounded as follows:

$$
\begin{aligned}
E(n, p) &= \sum_{k=0}^{n} \left[ p^{n-k} \cdot (1-p)^k \cdot \sum_{j=1}^{c_{n,k}} (S(n, k) + j) \right] \\
&= \sum_{k=0}^{\lceil n/2 \rceil} \left[ p^{n-k} \cdot (1-p)^k \cdot \sum_{j=1}^{c_{n,k}} (S(n, k) + j) \right] \\
&\quad + \sum_{k=\lceil n/2 \rceil+1}^{n} \left[ p^{n-k} \cdot (1-p)^k \cdot \sum_{j=1}^{c_{n,k}} (S(n, k) + j) \right] \\
&\leq O(f(\alpha^*)^n \cdot n^2) + d \cdot \left[ \sum_{j=1}^{c_{n,\lceil n/2 \rceil}} (S(n, k) + j) \right] \\
&\quad \cdot \sum_{k=\lceil n/2 \rceil+1}^{n} \left[ p^{n-k} \cdot (1-p)^k \right] \\
&\leq O(f(\alpha^*)^n \cdot n^2) + O(f(\alpha^*)^n \cdot n^2) \\
&= O(f(\alpha^*)^n \cdot n^2).
\end{aligned}
$$

In Lemma 1, we assumed that $Prob(x_i = 0) = p$ holds for all $x_i$. However, the probability is not usually the same. Therefore, we consider the case where $Prob(x_i = 0) = p + a_i$ holds for all $x_i$, where $a_i \geq 0$ $(i = 1, \ldots, n)$ and $p > \frac{1}{2}$. That is, we let $p = \min_{i=1}^{n} Prob(x_i = 0)$, where we can exchange the roles of $x_i = 0$ and $x_i = 1$ if $Prob(x_i = 0) < \frac{1}{2}$. Note that we are considering a situation that we have *a priori* information of some specific state in the target attractor and thus this assumption is reasonable.

**Theorem 1**. *Suppose that $x_i = b_i$ $(b_i \in \{0, 1\})$ holds with probability greater than or equal to p ($p > 0.5$) in one global state in the target attractor. Then, **ATTapriori** finds the target attractor using an $O([p^{1-\alpha}(1-p)^\alpha \beta^2]^n \cdot n^2)$ expected number of trials, where $\alpha = 1/(1 + \sqrt{p/(1-p)})$, $\beta = \left(\frac{1}{\alpha}\right)^\alpha \cdot \left(\frac{1}{1-\alpha}\right)^{1-\alpha}$.*

*Proof.* The proof strategy is to show that the expected number of trials in this setting is upper bounded by that in the case of $Prob(x_i = 0) = p$ for all $x_i$.

Let $p_i = Prob(x_i = 0) = p + a_i$ $(a_i \geq 0)$ and $D_P$ denote the corresponding probability distribution. Let **b** denote the step when **b** is examined, where **b** is a 0–1 vector of length $n$. Let $\mathbf{b}_i$ denote the $i$th element (*i.e.*, $i$th bit) of **b**. Let $\delta(x, y) = 0$ be the delta function: $\delta(x, y) = 1$ if $x = y$, otherwise $\delta(x, y) = 0$. Recall that **ATTapriori** examines bit vectors from those with a smaller number of 1s to a larger number of 1s, where ties can be broken in an arbitrary manner. For example, in the case of $n = 3$, we can examine in the order of 000,001,010,100,011,101,110,111, where we have $I_{000} = 1$, $I_{001} = 2$, $I_{010} = 3$, $\cdots$.

Then, the expected number of trials $E(n, D_P)$ is given by

$$
E(n, D_P) = \sum_{\mathbf{b}} \left( \prod_{i=1}^{n} (p + a_i)^{\delta(\mathbf{b}_i, 0)} (1 - (p + a_i))^{\delta(\mathbf{b}_i, 1)} \right) I_{\mathbf{b}},
$$

because the success probability for vector **b** is given by

$$
\prod_{i=1}^{n} (p + a_i)^{\delta(\mathbf{b}_i, 0)} (1 - (p + a_i))^{\delta(\mathbf{b}_i, 1)}
$$

and the number of trials done until the examination of **b** is $I_{\mathbf{b}}$. We prove the theorem (*i.e.*, $E(n, D_P) \leq E(n, p)$) by mathematical induction on the number of bits with $a_i > 0$.

In the base case, $p_i = p$ (i.e., $a_i = 0$) holds for all $i = 1, \ldots, n$. Therefore, the theorem follows from Lemma 1.

In the inductive step, we show that the expected number of trials does not decrease if we change one $p_i = p + a_i$ with $a_i > 0$ to $p_i = p$. Suppose that the theorem holds for any $D'_P$ in which $h$ nodes satisfy $p_i > p$. Let $D_P$ be a distribution in which $h + 1$ nodes satisfy $p_i > p$. We assume w.l.o.g. that $p_1 = p + a_1 > p$. Let $\hat{D}_P$ be the distribution that is obtained from $D_P$ by changing the value of $p_1$ to $p$. From the induction hypothesis, we have

$$E(n, \hat{D}_P) \leq E(n, p).$$

Let

$$p_{\mathbf{b}} = \prod_{i=1}^{n} (p + a_i)^{\delta(\mathbf{b}_i, 0)} (1 - (p + a_i))^{\delta(\mathbf{b}_i, 1)}.$$

Obviously, $E(n, \hat{D}_P)$ can be written as

$$E(n, \hat{D}_P) = \sum_{\mathbf{b}} p_{\mathbf{b}} I_{\mathbf{b}}.$$

Note that $p_1 = p$ is used in calculation of $p_{\mathbf{b}}$. Let $B_0$ (resp., $B_1$) be a set of 0–1 vectors $\mathbf{b}$ such that $\mathbf{b}_1 = 0$ (resp., $\mathbf{b}_1 = 1$).

Then, $E(n, D_P)$ is written as

$$
\begin{aligned}
E(n, D_P) \quad = \quad & \sum_{\mathbf{b} \in B_0} \left( \frac{p + a_1}{p} \right) \cdot p_{\mathbf{b}} \cdot I_{\mathbf{b}} \\
& + \sum_{\mathbf{b} \in B_1} \left( \frac{1 - (p + a_1)}{1 - p} \right) \cdot p_{\mathbf{b}} \cdot I_{\mathbf{b}},
\end{aligned}
$$

because prior probabilities $p$ and $1 - p$ for the first bit (i.e., $b_1$) in $\hat{D}_P$ are replaced by $p + a_1$ and $1 - (p + a_1)$ in $D_P$. Let $\mathbf{b}^{(0)}$ (resp., $\mathbf{b}^{(1)}$) be a bit vector obtained from $\mathbf{b}$ by letting $\mathbf{b}_1 = 0$ (resp., $\mathbf{b}_1 = 1$). Define $C_0$ and $C_1$ by

$$
\begin{aligned}
C_0 \quad &= \quad \sum_{\mathbf{b} \in B_0} p_{\mathbf{b}} \cdot I_{\mathbf{b}}, \\
C_1 \quad &= \quad \sum_{\mathbf{b} \in B_1} p_{\mathbf{b}} \cdot I_{\mathbf{b}}.
\end{aligned}
$$

Here we note that $I_{\mathbf{b}^{(0)}} < I_{\mathbf{b}^{(1)}}$ holds because the number of bits with 1 of $\mathbf{b}^{(0)}$ is smaller than that of $\mathbf{b}^{(1)}$. Since $\frac{1}{p} \cdot p_{\mathbf{b}^{(0)}} = \frac{1}{1-p} \cdot p_{\mathbf{b}^{(1)}}$ holds from the definition of $p_{\mathbf{b}}$, we have

$\left(\frac{1}{p}\right) \cdot C_0 < \left(\frac{1}{1-p}\right) \cdot C_1$. Therefore, we have

$$
\begin{aligned}
E(n, D_P) &= \left(\frac{p + a_1}{p}\right) C_0 + \left(\frac{1 - (p + a_1)}{1 - p}\right) C_1 \\
&= \left(1 + \frac{a_1}{p}\right) C_0 + \left(1 - \frac{a_1}{1 - p}\right) C_1 \\
&= a_1 \cdot \left[\frac{C_0}{p} - \frac{C_1}{1 - p}\right] + C_0 + C_1 \\
&\leq C_0 + C_1 \\
&= E(n, \hat{D}_P),
\end{aligned}
$$

which completes the mathematical induction.

It is to be noted that the expected time complexity of **ATTapriori** is $O([p^{1-\alpha}(1 - p)^\alpha \beta^2]^n \, poly(n))$ if each Boolean function can be evaluated in polynomial time and the length of each trajectory (including an attractor cycle) is polynomially bounded.

Since most practical Boolean functions can be evaluated in polynomial time and the lengths of trajectories in most practical BNs are considered to be not very long, this is a reasonable assumption.

Even if the trajectory is not bounded by a polynomial, we can modify the algorithm so that each trial is finished if the length of the trajectory exceeds some given steps (some polynomial steps). Since the proofs of Lemma 1 and Theorem 1 only discuss whether a given initial state is the same as the target state, this modification does not affect these theoretical results. It may affect the correctness of the original algorithm (i.e, the original algorithm may miss the desired attractor) because the period of the desired attractor may not be bounded by a specified polynomial, where the period of an attractor corresponds to a cyclic part of the trajectory (not the whole part of the trajectory). However, it is reasonable not to consider very long (e.g., exponentially long) attractors because any usual organism cannot live for exponentially long periods. For example, suppose that 1 step corresponds to 1 micro second (i.e., $10^{-6}$ second). Then, even for $n = 100$, $2^n$ is greater than $4 \times 10^{16}$ years. Limiting the length of the trajectory is also useful to reduce the space complexity. The algorithm need to memorize all states in the trajectory in order to find a cycle, which implies that the worst-case space complexity would be $O(2^n \cdot n)$ because the length of the trajectory can be $O(2^n)$ in the worst case. However, if we limit the length of the trajectory by some constant and it is enough to find one desired attractor, the space complexity would be $O(n)$.

In some cases, $x_1, \ldots, x_{n_1}$ satisfy that $Prob(x_i = 0) \geq p$ but there is no information on $x_{n_1+1}, \ldots, x_n$. In this case, for $k = 0, \ldots, n_1$, we examine $\binom{n_1}{k}$ assignments on $x_1, \ldots, x_{n_1}$, each of which has $k$ 1s. Furthermore, we examine $2^{n_2}$ assignments on $x_{n_1+1}, \ldots, x_n$ for each of $\binom{n_1}{k}$ assignments in the random order, where $n_2 = n - n_1$. Then, the resulting expected number of trials is

$$
\begin{aligned}
E(n, p) &= \sum_{k=0}^{n_1} \left[ p^{n_1-k} \cdot (1-p)^k \cdot \left(\frac{1}{2}\right)^{n_2} \cdot \sum_{j=1}^{2^{n_2} \cdot c_{n_1,k}} (2^{n_2} \cdot S(n_1, k) + j) \right] \\
&\leq \sum_{k=0}^{n_1} \left[ p^{n_1-k} \cdot (1-p)^k \cdot 2^{n_2} \cdot \sum_{j=1}^{c_{n_1,k}} (S(n_1, k) + j) \right].
\end{aligned}
$$

Since $\sum_{k=0}^{n_1} \left[ p^{n_1-k} \cdot (1-p)^k \cdot \sum_{j=1}^{c_{n_1,k}} (S(n_1, k) + j) \right]$ is $O(f(\alpha^*)^{n_1} \cdot n_1^2)$ from Lemma 1, this

number is

$$O(f(\alpha^*)^{n_1} \cdot 2^{n-n_1} \cdot n_1^2).$$

Suppose that $n_1 = cn$ holds for some constant $c$. Then, this number will be

$$O(f(\alpha^*)^{cn} \cdot 2^{(1-c)n} \cdot n^2).$$

Since $f(\alpha^*) < 2$, this number is $O((2 - \delta)^n n^2)$ for some constant $\delta$, where $\delta$ depends on $p$ and $c$.

The algorithm can be extended for the case in which there exist several levels of uncertainty (*i.e.*, we can use several $p_1, p_2, \ldots, p_h$ depending on node types). Note that the algorithm is deterministic and terminates once the desired attractor is found. Of course, how to define the desired attractor needs some discussions as discussed before. If we know some criteria to automatically decide whether or not a given attractor is a desired one, we can terminate the algorithm as soon as such an attractor is detected. Otherwise, the algorithm can be terminated if the number of trials exceeds a specified number or CPU time exceeds the time limit, where the number may be determined based on computational experiments as done below.

The **ATTapriori** algorithm is implemented in C, and the source code is available at https://github.com/takutsu5/AttPrior. In this version, at most 3 different values can be specified as probabilities (including probability 1.0) and bit vectors are examined from higher probabilities one to lower ones under a constraint that the number of changed bits for each probability class must not exceed the specified threshold.

## Analysis of synthetic random *N-K* networks

In order to evaluate the efficiency of the proposed algorithm, we performed computational experiments to detect attractors using synthetic random *N-K* networks, where *N* and *K* indicate the number of nodes and the indegree, respectively, and the update function of each node is controlled by exactly *K* nodes which are uniformly randomly selected. In this section, random *N-K* networks were generated using the *generateRandomNKnetwork* function provided by the R-package *BoolNet* [27].

First, we measured the average number of trials of 1000 experiments until a bit vector of global state examined by **ATTapriori** algorithm corresponds with the target attractor. In this experiment, random *N-K* networks of $N$ = 5, 10, 15, 20, 25, 30, 35, 40, and $K$ = 2 were generated, and one of the attractors detected by the *getAttractor* function of *BoolNet* was set as the target attractor for each network. In order to avoid matching with the target attractors in the first trials, the bit vectors generated by changing each bit of the target attractors according to the prior probability are given as the initial states.

We assigned the same prior probability $p$ to all nodes, and changed $p$ from 0.7 to 0.9 in steps of 0.05, and then the upper bound of the expected number of trials given by Theorem 1 was calculated for each $p$.

The results of comparing the average number of trials with the theoretical value using random *N-K* networks are shown in Fig 2A. The average number of trials increases exponentially with the increase of the network size *N*, and is upper-bounded by the expected number of trials obtained theoretically. On the other hand, the average number of trials decreases as the prior probability increases, and $3.50 \times 10^{10}$ and $9.62 \times 10^6$ trials are required when the prior probabilities are 0.7 and 0.9 in the case of $N$ = 40, which indicates that the higher prior probability enables us to detect attractors much faster than without *a priori* information.

Second, we performed another experiment to show the distribution of the lengths of trajectories until reaching attractors. Fig 2B indicates the average length of trajectories of 100 experiments from random initial states to attractors without period lengths for random *N-K*

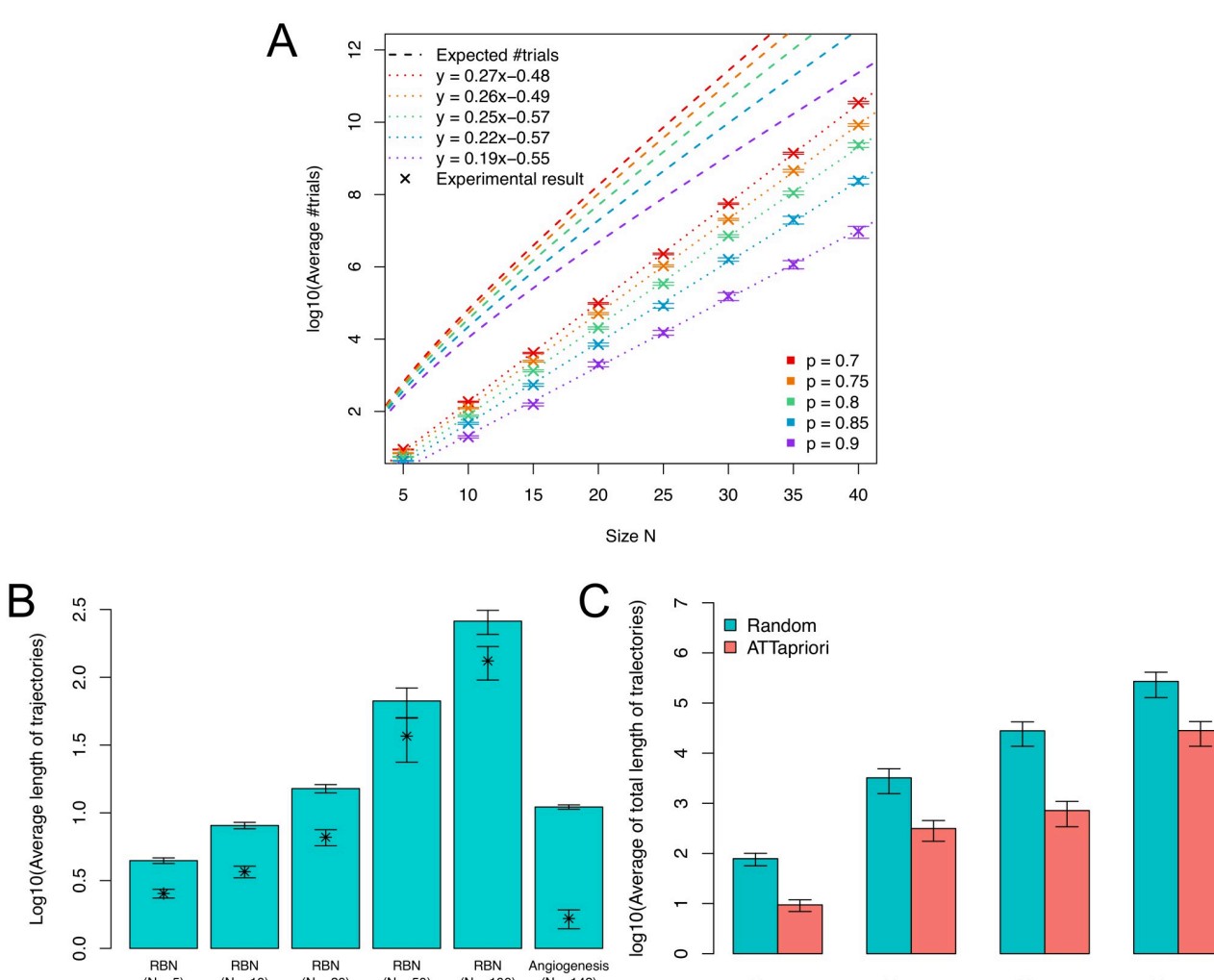

**Fig 2. Analysis of synthetic random networks and the angiogenesis network.** (A) The number of expected trials in synthetic random networks. The average number of trials and the expected number of trials for attractor detection of synthetic random *N-K* networks with the prior probability from 0.7 to 0.9. The horizontal axis shows the size of input random *N-K* networks, where *K* is fixed at 2. The dotted lines indicate the regression lines for the experimental results. (B) The lengths of trajectories of synthetic random networks (RBNs) and the angiogenesis network. The average lengths of trajectories of 100 trials from random initial states to attractors for random *N-K* networks (*N* = 5, 10, 20, 50, 100 and *K* = 2) and the angiogenesis network (*N* = 142). The symbol '*' and the bars indicate the averages of period lengths and the standard errors, respectively. (C) The total lengths of trajectories required to reach the target attractors. The **ATTapriori** and **Random** are the cases with and without *a priori* information, respectively. The result shows that the average total lengths of trajectories of 100 experiments not including period lengths until reaching the target attractors for the BN sizes of *N* = 10, 20, 30, and 40, where the prior probability in **ATTapriori** was set to 0.90.

networks whose sizes are *N* = 5, 10, 20, 50, 100 and indegrees are *K* = 2. As a real network, the lengths of trajectories of the angiogenesis network of size *N* = 142 was also measured (its details are described in the next section) [9]. The symbol '*' indicates the averages of period lengths. Although the average lengths of trajectories becomes longer as the size gets larger, it is less than 300 even when the size of network is 100. In addition, the average length of trajectories of the angiogenesis network is much shorter than that of random networks of size *N* = 100.

Finally, we compared the total lengths of trajectories until reaching a target attractor with (**ATTapriori**) and without (**Random**) *a priori* information. In this experiment, a target attractor was set in advance, which was detected by *BoolNet* package of R. The **Random**

method starts from a random initial state and repeats state transitions until an attractor is found. If the found attractor is the target attractor, it outputs the total lengths of trajectories and terminates. Otherwise, it starts from another random initial state and repeats the above procedure. On the other hand, in **ATTapriori**, the same procedure was done except that the initial states are decided by **ATTapriori** algorithm. Fig 2C shows that the average total lengths of trajectories of 100 experiments not including period lengths until reaching the target attractors for the sizes of $N$ = 10, 20, 30, and 40, where the prior probability in **ATTapriori** was set to 0.90. From the results, the average total length of trajectories of **ATTapriori** was shortened to about 1/40 at the maximum. Additional information and files can be found in S2 File.

## Effect of microenvironment on EC behavior revisited (BN with $n$ = 142 nodes)

We next applied the proposed algorithm to the analysis of a BN describing the effect of several microenvironments on EC behavior during angiogenesis [9]. The authors of the original study developed two BN versions: a full BN with $n$ = 142 nodes which we analyze here, and a reduced BN with $n$ = 64 nodes, which was analyzed in the original study. The two BNs are formulated in the classical way, where each node corresponds to a molecular entity or a conceptual place-holder (such as *shear stress*). Using the reduced model, the authors identified in total 67 micro-environments of which 35 induce typical, and 32 atypical EC behavior. The authors used three signatures of four molecular markers (see Introduction; Materials and methods) to decide which EC behavior (tip, stalk, or phalanx) was triggered by each microenvironment. A micro-environment induces typical behavior if all detected attractors show an EC marker configuration corresponding to a single and stable EC signature. Otherwise, the microenvironment is regarded as inducing atypical behavior. For example, if the EC marker signature does not correspond to any of the three behaviors, the induced behavior is regarded as atypical. Here, we investigated the attractor landscape in the full model with $n$ = 142 nodes as provided by the authors [9]. We use the same definition as the authors to interpret EC behavior based on the signature of the four EC markers, and use the same microenvironments inducing a specific behavior. (*I.e.*, microenvironment numbers 1–2, 3–16, and 17–35 induce phalanx, stalk, and tip, respectively; numbers 36–67 induce atypical behavior). Furthermore, our analysis requires initial guesses for each microenvironment, and *a priori* probabilities. We base our initial guesses on the in total 67 microenvironment configurations by the authors of the original study [9], which each comprise 16 nodes. The authors also include 10 nodes with constitutive activity, which we also incorporate in our initial guess. Hence, in total, we can assign 26 node values to our initial guess based on the original study, and assume that their *a priori* probability is 1 (Materials and methods). Since the initial values of all of the remaining 116 nodes are not reported in the original study [9], we test two scenarios: one in which they are set to 0 (OFF scenario), and one in which they are set to 1 (ON scenario). We call this type of guess with partially known, and partially unknown initial values a semi-informed guess. For these remaining nodes, we test for each of these scenarios (ON and OFF) four arbitrarily chosen *a priori* probabilities: 0.7, 0.8, 0.9, and 1 (Materials and methods).

First, our attractor analysis shows that for the microenvironments inducing typical behavior (numbers 1 to 35), point attractors or periodic attractors with period 2 can be identified (Fig 3 and Fig A in S1 File). The detected periodic attractors for the microenvironments (numbers 36 to 67) inducing atypical behavior show higher variation in their lengths, with *per* = 2, 4, 6, 8, 18 (Fig A in S1 File).

Second, we confirm that the 33 microenvironments (numbers 3 to 35) previously found to induce either stalk or tip behavior induce the expected behavior in the full network (Fig 3).

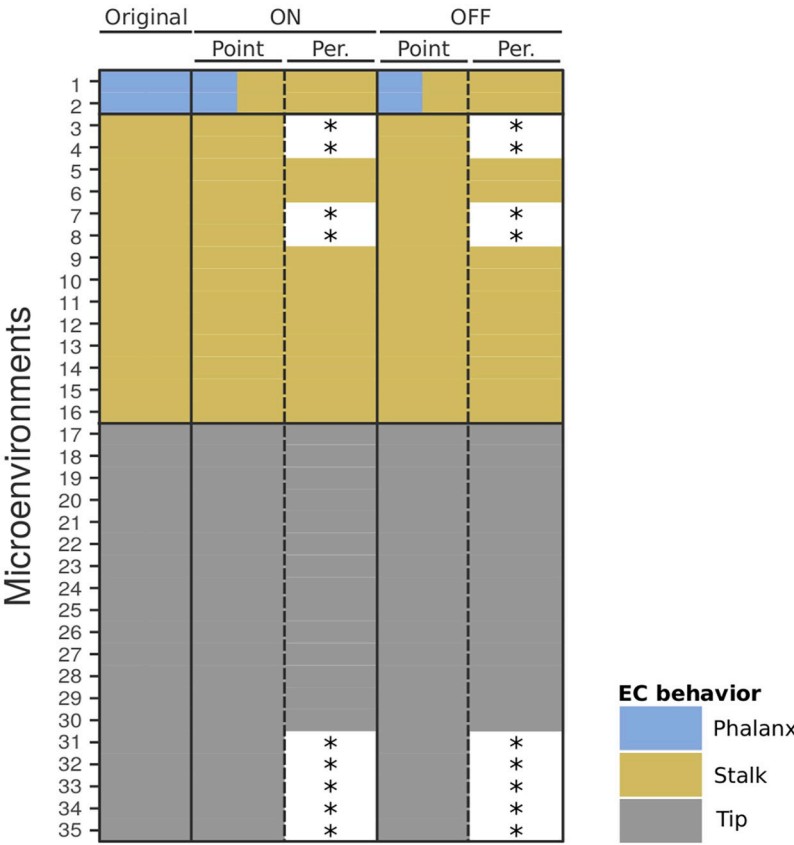

**Fig 3. Typical endothelial cell (EC) behaviors induced by 35 microenvironments.** Comparison of the induced EC behavior from 35 microenvironments (rows) to the results from the original study [9], *Original* column, with our analysis, *ON* and *OFF* columns. In the underlying BN three types of EC behavior, phalanx, stalk, and tip, were predicted to be induced dependent on the microenvironment. In the original study [9] based on a BN with $n = 64$ nodes (*Original*), microenvironments 1–2 induce phalanx behavior (blue), 3–16 induce stalk behavior (yellow), and 17–35 induce tip behavior (grey). We used the full network with $n = 142$ nodes for our analysis requiring an initial guess and *a priori* probabilities. We could assign 26 node values based on the original study, and tested two scenarios for the remaining 116 nodes as our initial guess: the ON scenario, in which the remaining nodes are set to 1 (*ON* column), and the OFF scenario (*OFF* column), in which the remaining nodes are set to 0. Results using *a priori* probability 0.7 shown. For both types of guesses (ON and OFF scenario), both point, and periodic attractors with period 2 were detected. Per.: periodic. *No periodic attractors detected. (The same microenvironment numbering is applied as in the original study).

Similarly, all 32 microenvironments (numbers 36 to 67) identified to induce atypical behavior also induce atypical behavior in the full network (Fig B in S1 File).

Third, the two microenvironments (numbers 1, 2) previously described to induce only phalanx behavior were found to induce two types of behavior: phalanx and stalk behavior (Fig 3). This finding indicates that the conclusion about the two previously defined microenvironments (numbers 1,2) in the reduced network does not directly apply to the full network.

**Analysis of a cell cycle control network (bBM with $n = 3158$ nodes).** We next considered a cell cycle control network of *S. cerevisiae* [28] formulated as a bBM with 3158 nodes. Due to the bBM formalism, a few considerations need to be taken for the attractor analysis (see Introduction). The **ATTapriori** algorithm may find attractors in which two or more mutually exclusive states (*e.g.*, DNA both replicating and replicated at the same time) are simultaneously true, or a necessary component is not true. While these attractors are technically possible, they do not convey any biological meaning. We regard these attractors as technical artefacts and

refer to them as *invalid*. In addition, the algorithm may remove a gene or the last true state of a specific site, effectively creating a deletion mutant of the corresponding component(s) and hence altering the genotype of the model. Next, we regard all other attractors to correspond to a biologically relevant genotype, with either of two possible corresponding phenotypes: viable or lethal. We regard an attractor to correspond to a viable phenotype if all macroscopic processes are traversed, as this would be required for a cell to successfully duplicate. This is only possible for periodic attractors. Similarly, we regard an attractor to correspond to a lethal phenotype if the attractor is a point attractor, or a periodic attractor which does not traverse all macroscopic states. Such a lethal phenotype may correspond to a mutant in which an essential gene is missing.

First, we studied the attractors of the genotype corresponding to the wildtype phenotype where we used the previously known periodic attractor as initial guess. We regard this an informed guess, where we also set two necessary network inputs *Nutrients, Histones* to 1, and four other inputs, corresponding to chemical inhibitors of the cell cycle, to 0. We tested four different, arbitrarily chosen *a priori* probabilities, 0.7, 0.8, 0.9, and 1 in different combinations, except for the in total six inputs, where the *a priori* probabilities are all set to 1 (see Materials and methods). We discovered 66 unique periodic attractors with period 186, and 125 unique point attractors (Fig 4). We then analyzed these 66 unique periodic attractors (Table A in S1 File) by manually scanning for nodes that are constitutively 0, and compare this to the wildtype attractor. If the constitutively inactive states all belong to one gene, we consider this gene to be absent in the attractor. We found that one attractor corresponds to the wildtype, 39 correspond to viable deletion mutants (gene-specific sites at The Saccharomyces Genome Database

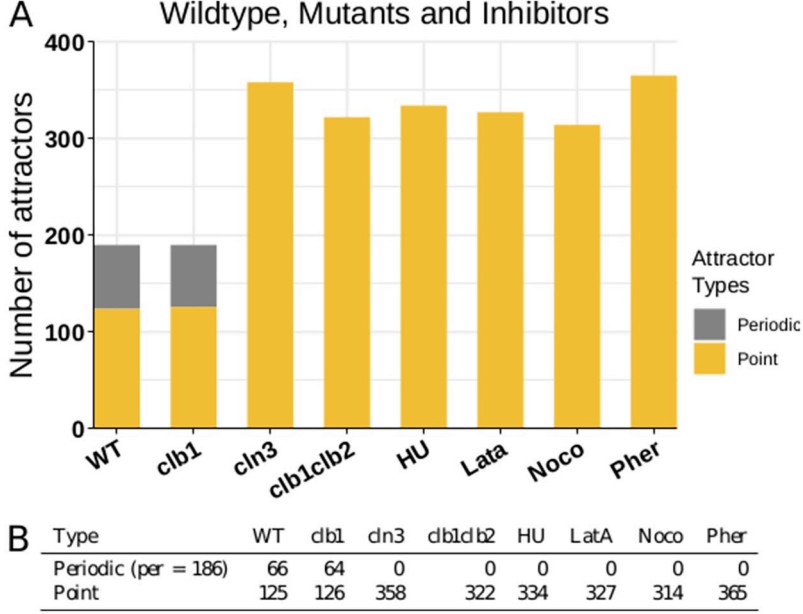

| Type | WT | clb1 | cln3 | clb1clb2 | HU | LatA | Noco | Pher |
|---|---|---|---|---|---|---|---|---|
| Periodic (per = 186) | 66 | 64 | 0 | 0 | 0 | 0 | 0 | 0 |
| Point | 125 | 126 | 358 | 322 | 334 | 327 | 314 | 365 |

**Fig 4. Attractor analysis results of cell cycle control network.** Attractor analysis result of a bBM cell cycle control network with *n* = 3158 nodes. Results from in total eight initial guesses are shown: the wildtype genotype, the *clb1* genotype with a corresponding viable phenotype, the *cln3* genotype with a corresponding lethal phenotype (due to absence of Bcks2 in the underlying model), the *clb1clb2* genotype with a lethal corresponding phenotype, and four initial guesses where one of the following cell cycle inhibitors is active, hydroxyurea, latrunculin, nocodazole, and pheromone. Several combiantions of *a priori* probabilities were used, representative results shown. (A) Barplot of the found types of attractors (point or periodic) for the eight initial guesses. (B) Data table. WT: wildtype, HU: hydroxyurea, Lata: Latrunculin A, Noco: nocodazole, Pher: Pheromone, per: period.

(SGD) [31]), two correspond to deletion mutants with conflicting reported phenotypes (PP2A, Net1), and 16 to deletion of hypothetical kinases and phosphatases introduced in the model construction process (not testable). Only one corresponds to a lethal deletion mutant; *sec2*, where the essential function is not part of the model. The remaining 7 are technical artefacts, where an infeasible pattern of macroscopic states was active in the initial perturbation. Hence, all the newly discovered periodic attractors correspond either to deletion mutants or to technical artefacts, and no new basins of attractions in the wildtype were discovered. The 125 unique point attractors (Table B in S1 File) can by definition only correspond to lethal phenotypes. We manually examined the attractors for inactive nodes, and assigned lethal mutations if the inactive nodes corresponded to one or several genes. We found that 71 of the 125 detected point attractors indeed correspond to known lethal mutants. 28 mutants have a viable phenotype *in vivo*, but redundancy mechanisms that compensate for loss of their functions are not included in the model. 18 mutants showed a missing structural component, and hence, the phenotype is lethal (model phenomenon, technical artefact). The remaining eight attractors constitute technical and other mutants which could not be clearly categorized, or where the reported *in vivo* phenotype is conflicting.

Fig 4 also shows the attractors obtained using other initial guesses: the viable *clb1* mutant phenotype, and the two lethal mutant phenotypes *cln3* (due to the absence of Bck2) and *clb1clb2*.

Second, we perturbed the network with the four cell cycle inhibitors hydroxyurea (HU), Latrunculin A (LatA), nocodazole (Noco), and pheromone (Pher), and analyzed the attractors. Perturbing the cell cycle network with either of the four inhibitors arrests cell cycle progress and hence, we expect point attractors. Fig 4 shows that the inhibitory effect is reflected in the identified attractors, as only point attractors were observed, corresponding to lethal phenotypes.

Third, we introduced two types of functional mutations: AND to OR mutations (functional promiscuity), and OR to AND mutations (functional restriction), with mutation rates of 0.1%, 1%, and 10%, and calculated the attractors. The two types of functional mutations promote different kinds of attractors with increasing mutation rates: Functional promiscuity mutations promote periodic attractors (Fig C in S1 File), whereas functional restriction mutations promote point attractors (Fig D in S1 File). Overall, the periodic attractors had a high variation, with lengths of 2, 3, 4, and 32, and none of them was found to correspond to a viable phenotype. Furthermore, with increasing mutation rate, the detected attractors did no longer correspond to a biologically meaningful pattern and are regarded as invalid. In comparison, in the wildtype and *clb1* mutant (Fig 4), periodic attractors corresponding to viable phenotypes were found. The majority of the point attractors detected in the *cln3* and *clb1clb2* mutants have a biologically valid signature, and correspond to lethal phenotypes. We observe that mutations in the Boolean functions with rates as low as 0.1% induce lethal behavior.

Finally, we tested random initial states as initial guesses. Our previously known periodic attractor corresponding to the wildtype is an informed guess about an existing periodic and biologically relevant attractor. Here, we used 10 random initial states to test if our algorithm can detect periodic and/or biologically relevant attractors with random, and hence, uninformed guesses. We tested three scenarios, one with completely random initial guesses, one in which the input *Nutrients* was set to on, and one in which *Nutrients* was set to off. In all three scenarios, only point attractors were detected (Fig E in S1 File).

## Discussion

In this study, we addressed the attractor detection problem in deterministic BNs. We have shown that *a priori* information is useful for reducing the computational complexity of the

attractor detection problem, both theoretically, and by applying it to two biological BNs. Note that it is almost impossible to detect attractors in these two biological BNs without *a priori* information because the size of the state space is quite large ($2^n$ for $n = 142$ and $n = 3158$).

Our analysis of the angiogenesis network [9] with 142 nodes showed that two previously defined microenvironments induce, contrary to earlier findings, two behaviors, phalanx and stalk. This observation stresses an important aspect: The conservation of network properties in a reduced network with asynchronous updates [10] may not be applicable for networks with deterministic updates. We suggest a refinement of the phalanx inducing microenvironment with a possible role for glycolytic activity (reviewed in [32]).

A previous study of cell cycle control using a highly simplified network [33] showed that the $G_1$ state is highly stable with a huge basin of attraction, and a point attractor after traversing the cell cycle once. We were able, for the first time, to perform a heuristic attractor analysis of a mechanistically detailed cell cycle control network with 3158 nodes [28]. Due to its size, we cannot calculate the basin of attraction, but expand on the view of robustness. Here, the method of checking for alternative basins of attraction also has a chance to introduce *de facto* deletion mutants by removing components. We do indeed see that virtually all new attractors correspond to mutant forms of the network. We find two types of attractors, *periodic* and *point*, corresponding to a viable or lethal phenotype, respectively. The viable characteristics are conserved upon mutations of functionally redundant or non-essential genes. Mutations of essential genes show point attractors, corresponding to a lethal phenotype. Hence, network dynamics interpreted as cell viability are conserved upon perturbations such as mutations. Perturbations altering gene functionality of the whole network induced by changing the underlying Boolean rules promote *periodic* attractors with no viable corresponding phenotype in networks with increasing functional promiscuity, and *point* attractors with no viable corresponding phenotype in networks with increasing functional restriction. This means that strong perturbations influencing the whole network generally lead to cell inviability, and not to erroneous multiplication of sick phenotypes.

The attractor analysis of the cell cycle control bBM revealed another benefit of the proposed algorithm: With a given periodic attractor and *a priori* information, the algorithm detected mutational genotypes with a directly interpretable corresponding phenotype based on the type of detected attractor. The bBM formalism requires to disable several nodes beloning to a certain component in order to mimic a genotype with a gene knock out. The analysis of our attractors revealed that the **ATTapriori** alrogithm automatically removed the nodes to produce a certain knock out. That is, the algorithm automatically performed an *in silico* mutagenesis screen, a task which had to be performed manually previously. Due to the expressiveness of bBMs, a mutational analysis requires more than removing a single node to knockout a gene. Moreover, the resulting attractors are directly interpretable, assisting a fast verification of known mutants, and potentially detecting new phenotypes.

We show that we were able to break the $O(2^n)$ barrier for the detection of periodic attractors by using *a priori* information. An important step is choosing an initial guess to which we assign the *a priori* information. The precision of this initial guess to be part of a periodic attractor becomes more crucial with increasing network size. In some cases, *a priori* information can be obtained from binarized gene expression data. However, most often, this kind of data is not available. Nevertheless, our analyses show that *a priori* information can be chosen arbitrarily. We could identify periodic attractors in the angiogenesis BN with 142 nodes using a semi-informed initial guess. For the cell cycle network, however, a previously known periodic attractor was necessary to discover additional periodic attractors, as random initial guesses were not sufficient. This means that the problem of initially finding a periodic attractor with certain properties may not be conveniently approached with our proposed algorithm for

larger networks. A systematic study to investigate this phenomenon thoroughly would be required. Additionally, due to the network size (3158 nodes) and the relatively small space which we can search, we cannot rule out the possibility that other periodic attractors exist. Furthermore, the proposed algorithm still needs $O(2^n poly(n))$ time if there is no *a priori* information (*i.e.*, $p = 0.5$). Therefore, actually breaking the $O(2^n)$ barrier is left as an open problem.

Another direction of future studies on detecting periodic attractors is to combine the idea of using *a priori* information with practical solvers for NP-hard problems such as ILP and SAT solvers. Of course, as mentioned in the Introduction, these solvers may not be directly applied to detection of long attractors. However, these might be effectively applied if *a priori* information is utilized to reduce the search space. In addition, *a priori* information might be useful to efficiently stabilize biological systems [7, 8], where a BN is called *stabilizable* if there exists a feedback control and a point attractor such that the BN can be driven to the attractor beginning from any state. *A priori* information might be utilized to determine such a point attractor. Therefore, it would be worthy to study these directions. Although we have considered synchronous BNs in this work, various models of asynchronous BNs have been proposed and studied [19, 20]. Unfortunately, our approach cannot be applied to such BNs because trajectories are not uniquely determined. Therefore, it is interesting to study how *a priori* information can be utilized for attractor detection in asynchronous BNs.

In summary, BNs are a versatile tool to describe biological processes, and have recently been developed to express mechanistic detail in signal transduction pathways [29] using a bipartite structure. bBMs overcome the parameter estimation problem associated with large-scale modeling efforts while allowing a mechanistic description of a biological process. Thus, although bBMs potentially pave the way towards the development of mechanistic, whole-cell models, they are associated with the challenge of the attractor detection problem [17]. In this paper we demonstrate that the proposed algorithm can be meaningfully applied to heuristically explore the attractor space with periodic attractors of various lengths to identify attractors in biological BNs.

## Materials and methods

### BN analysis

**Angiogenesis network analysis.** We consider the originally proposed full angiogenesis BN [9] consisting of $n = 142$ nodes for our analysis. Previously, a reduced version with $n = 64$ nodes was considered for analysis in the original study [9]. Our analysis requires an initial guess and *a priori* probabilities which we base as much as possible on the original study. In the original study, 16 nodes were identified to comprise the microenvironment (*VEGFC Dp, VEGFAxxxP, ANG1, Oxygen, ShearStress, JAGp, DLL4p, WNT5a, WNT7a, FGF, IGF, BMP9, BMP10, TGFB1, VEGFC D, AMPATP*). In total, 67 patterns of activation reflecting 67 different microenvironments were identified, which we used for our analysis. Furthermore, 10 nodes with a fixed value were used (*ACVR2A, BMPRII, TGFBRII, sGC, SMAD4, γ-Secretase, ADAM10/17* with value 1; *Ryk (sFRP1), DKK1/3, BTrCP* with value 0). We assigned an *a priori* probability of 1 to each of these initial guesses.

For the remaining 116 nodes, we tested two scenarios: The ON scenario, in which all remaining nodes are set to 1 in the initial guesses, and the OFF scenario, in which all remaining nodes are set to 0 in the initial guesses. These nodes were assigned the same *a priori* probability, where we tested in total four values (0.7, 0.8, 0.9, and 1.0) for each of the two scenarios. Hence, for each of the 67 microenvironments, the two scenarios (ON and OFF) were tested with four different *a priori* probabilities, adding up to in total 536 tests.

We analyzed the detected attractors in terms of EC behavior. EC behavior was assigned according to the patterns of activity of marker nodes as assigned in the original study [9]: Phalanx: $AKT = 1$, $JAGa = 0$, $NRP1 = 0$; Stalk: $JAGa = 1$, $NRP1 = 0$; Tip: $NRP1 = 1$, $DLL4a = 1$, $AKT = 0$. Additional information and files can be found in S3 File.

**Cell cycle control network analysis.** The cell cycle control bBM [28] describes the control and regulation of three macroscopic cellular processes required for cell cycle progression: DNA replication, spindle pole body duplication, and cell growth. In the model, each macroscopic process traverses a set of irreversible biological states, which allows monitoring of the cell state and cell cycle progression. As initial guess we used an informed guess, a single state from a previously known attractor with period 186 corresponding to the wildtype. The bBM uses *Nutrients* and *Histones* as an input, and can be perturbed with in total four cell cycle inhibitors. The corresponding nodes in the initial guesses were set to 1 and 0, respectively, and their *a priori* probabilities set to 1, except for the scenarios in which the cell cycle inhibitors were explicitly tested. Furthermore, the underlying bBM has two types of nodes, one for reactions, and one for states. For all reaction nodes, and for all state nodes, the same probabilities were used, respectively. Pairwise combinations of four *a priori* probabilities (0.7, 0.8, 0.9, and 1.0) were tested, resulting in 16 different scenarios for each network. From these 16 scenarios, the one with the highest number of detected attractors was used for analysis. The proposed algorithm perturbs the initial guess. Due to the architecture of the bBM, this does not only perturb the starting vector, but also has a chance of permanently removing one or more components, *de facto* creating a deletion mutant. Hence, a small fraction of the detected attractors corresponds to technical artefacts, *e.g.* if none or several of the mutually exclusive macroscopic cell cycle states were initiated as true. These artefacts were omitted from the following analysis. We evaluated if a newly detected attractor corresponds to a viable phenotype by checking i) if at each time point, exactly one macroscopic state is true in each macroscopic process, ii) if each macroscopic process traverses through all its states. From this follows that only periodic attractors potentially correspond to a viable phenotype. Point attractors may correspond to a lethal phenotype induced by one or a combinations of lethal mutations. This happens if all complementary states for a particular site are set to 0 (inactive) for an essential component whose turnover (synthesis) is not considered in the model (see Introduction), *de facto* creating a deletion mutant. Similarly, if a macroscopic state is turned on or off, it is very likely to introduce an unfeasible state where none or more than one of these mutually exclusive states are 1 (active).

Four genotypes were tested: wildtype, two single knockout mutants (*clb1* and *cln3* deletions), and a double knockout mutant (*clb1clb2* deletion). The initial guess for the mutants was based on the initial guess for the wildtype so that the reactions and states of the genes to be deleted were set to zero, and *a priori* probability set to 1. Additionally, two types of functionally mutated networks of these four scenarios were used. In the case of functional promiscuity, AND operators were changed to OR. In the case of functional loss, OR operators were changed to AND. For both versions three mutation rates 0.1%, 1% and 10% with uniformly distributed probabilities were used, summing up to six versions for each of the four initially tested networks. The corresponding initial guesses were used. The two network inputs *Nutrients* and *Histones* were active and their *a priori* probabilities set to 1. The four cell cycle inhibitors were off and their *a priori* probabilities set to 1, except for the versions where the influence of these were tested one by one.

The random initial guesses were created by assigning values of 0 or 1 to all nodes in the network following a binomial distribution. All attractor searches were conducted so that the **ATTapriori** algorithm was run until a CPU time limit was exceeded. Additional information and files can be found in S3 File.

## Supporting information

**S1 File. This file contains Figs A-E, and Tables A-B. Fig A. Attractors of typical and atypical behavior inducing microenvironments**. In the original study [9], the authors identified 35 microenvironments which induce typical endothelial cell (EC) behavior, and 32 microenvironments which induce atypical EC, behavior using a reduced network with $n = 64$ nodes. We show the types of attractors (point, periodic, and lengths of periodic attractors) from our analysis of the full network with $n = 142$ nodes. As initial guesses, the previously defined values for each microenvironment comprising 16 nodes were applied. Additionally, 10 nodes with a fixed value were used. For the remaining 116 nodes, two scenarios were tested, one in which all remaining nodes are set to 1 (ON scenario), and one where all remaining nodes are set to 0 (OFF scenario). As *a priori* information, the probabilities 0,7, 0.8, 0.9, and 1 were tested. Hence, for each microenvironment and ON or OFF setting, four tests were performed. For each number and scenario, the same results were retrieved (exception for 1). The results for *a priori* probability 0.7 are shown. *Left*: Attractor distributions for microenvironments (numbers 1–35) inducing typical behavior. *Right*: Attractor distributions for microenvironments inducing atypical (numbers 36–67) behavior. **Fig B. Endothelial cell (EC) behavior of atypical behavior inducing microenvironments**. In the original study [9], the authors identified 32 microenvironments predicted to induce atypical EC behavior. EC behavior is interpreted by the signature of four markers (*AKT1*, autocrine *JAG1*, *DLL4a*, and *NRP1*). A microenvironment induces atypical EC behavior if the EC marker signature does not correspond to phalanx, stalk, or tip, or if their signature is not stable in the detected attractors. We based our initial guesses required for attractor analysis on the information provided in the original study, from where we can assign node values for 16 nodes comprising the microenvironment, and 10 nodes with fixed values. For the remaining 116 out of 142 nodes, we tested two scenarios, one in which all remaining nodes are set to 1 (ON scenario), and one where all remaining nodes are set to 0 (OFF scenario). Shown are the EC behaviors interpreted from the detected attractors from our analysis. Rows: Microenvironment numbers (numbers 36–67 corresponding to the original numbering) predicted to induce atypical EC behavior. *ON* column: Results from scenarios where the unkown node values in the initial guess were set to 1 (ON). *OFF* column: Results from scenarios where the unkown node values in the initial guess were set to 0 (OFF). Rows (microenvironments) which resulted in attractors corresponding to only one EC behavior (numbers 36–38, and 48–59) showed an instable EC marker signature, and are regarded to induce atypical behavior. **Fig C. Attractors of functionally promiscuous mutants**. Results from the attractor analysis of a cell cycle control network with $n = 3158$ nodes. The Boolean rules in the original network were mutated from AND to OR, mimicking a functionally promiscuous mutant. Three networks were generated with mutation rates of 0.1%, 1%, 10%. For each network, four initial guesses were used, the wildtype genotype, and the *clb1*, *cln3*, and *clb1clb2* mutant genotypes. The detected attractors were interpreted in terms of validity, and viability of the corresponding phenotype. Due to the bipartite Boolean modeling formalism, our proposed algorithm may detect attractors where two or more mutually exclusive nodes are active, or essential components are inactive. While these types of attractors are technically possible to detect, they do not carry biological meaning and we refer to them as *invalid*. The remaining attractors are regarded as biologically valid attractors, with two possible corresponding phenotypes: *viable* and *lethal*. (A) Detected attractors using four initial guesses, and three mutational rates which affect the Boolean rules. (B) Data table. WT: wildtype, Per.: Periodic, Inval.: Invalid. **Fig D. Attractors of functionally restricted mutants**. Results from the attractor analysis of a cell cycle control network with $n = 3158$ nodes. The Boolean rules in the original network were mutated from OR to AND, mimicking a functionally restricted mutant.

Three networks were generated with mutation rates of 0.1%, 1%, 10%. For each network, four initial guesses were used, the wildtype genotype, and the *clb1*, *cln3*, and *clb1clb2* mutant genotypes. The detected attractors were interpreted in terms of validity, and viability of the corresponding phenotype. Due to the bipartite Boolean modeling formalism, our proposed algorithm may detect attractors where two or more mutually exclusive nodes are active, or essential components are inactive. While these types of attractors are technically possible to detect, they do not carry biological meaning and we refer to them as *invalid*. The remaining attractors are regarded as biologically valid attractors, with two possible corresponding phenotypes: *viable* and *lethal*. (A) Detected attractors using four initial guesses, and three mutational rates which affect the Boolean rules. (B) Data table. WT: wildtype, Per.: Periodic, Inval.: Invalid. **Fig E. Attractors using either a known periodic attractor or random initial states in the cell cycle control network**. Results from the attractor analysis of a cell cycle control network with $n$ = 3158 nodes using either the wildtype genotype as initial guess (upper panel), or in total 10 random initial guesses (second panel from above). Additionally, these 10 random initial guesses were in one scenario modified so that the network input *Nutrients* was set to 0 (OFF scenario, third panel from above), and in another scenario so that the network input *Nutrients* was set to 1 (ON scenario, bottom panel). Furthermore, in the bipartite network, there are nodes corresponding to biochemical reactions, and nodes corrsponding to elemental states of the components included, resulting in two node types. For each node type, the same *a priori* probability was used (0.7, 0.8, 0.9, or 1.0), and the pairwise combinations of the *a priori* probabilities between the two node types tested. The y-axis labels indicate the *a priori* probabilities for reaction and state nodes. Results shown for random initial state 1, respectively. p: period; pr: probability. **Table A. Periodic attractors and their corresponding phenotypes using wildtype as initial guess (related to** Fig 4, **main manuscript)**. *Viable*: correctly identified viable genotypes; *not testable*: cellular function where corresponding gene product not identified. Parenthesis: Number of attractors. Not shown: wildtype attractor (1), technical (7), *sec2* (1, lethal), *net1*, *pp2a* (2, contradictory). *Technical*: initial conditions included none or more than one of the mutual exclusive cell cycle stages; *lethal*: incorrectly identified as viable, true phenotype lethal; *contradictory*: contradictory statements reported in literature. All found attractors are unique. Viability status from The Saccharomyces Genome Database (SGD) [31].
**Table B. Point attractors and their corresponding phenotypes using wildtype as initial guess (related to** Fig 4, **main manuscript)**. *Essential*: Correctly identified as lethal mutants; *non-essential*: incorrectly identified as lethal, paralogs with compensating function *in vivo* not included in model; *structural*: nodes in network corresponding to structural compounds (*e.g.* polymerase II) not modelled as individual gene products or corresponding to a cellular state (*e.g.* ssDNA); *not testable*: cellular function where corresponding gene product not identified, (*) contradictory statements reported in the literature. Parenthesis: Number of detected attractors. Double mutants with *cdh1* were found for attractors occurring twice. Not shown: *technical* where initial conditions included none or more than one of the mutual exclusive cell cycle stages (2); attractors where genotype could not be identified (2). All found attractors are unique. Viability status from SGD [31].
(PDF)

**S2 File. A zip archive containing code to run computational experiments of synthetic random Boolean networks and the angiogenesis network. Instructions can be found in the README.txt within the archive.**
(ZIP)

**S3 File. A zip archive containing auxiliary files to retrieve attractors from the angiogenesis and cell cycle networks. Instructions can be found in the README.txt within the archive.**
(ZIP)

## Author Contributions

**Conceptualization:** Tatsuya Akutsu.

**Data curation:** Ulrike Münzner.

**Formal analysis:** Tatsuya Akutsu.

**Funding acquisition:** Ulrike Münzner, Tatsuya Akutsu.

**Investigation:** Ulrike Münzner, Tomoya Mori, Marcus Krantz, Edda Klipp, Tatsuya Akutsu.

**Methodology:** Ulrike Münzner, Tomoya Mori, Tatsuya Akutsu.

**Resources:** Ulrike Münzner.

**Software:** Ulrike Münzner, Tomoya Mori, Tatsuya Akutsu.

**Supervision:** Tatsuya Akutsu.

**Validation:** Marcus Krantz, Edda Klipp.

**Visualization:** Ulrike Münzner, Tomoya Mori.

**Writing – original draft:** Ulrike Münzner, Tomoya Mori, Tatsuya Akutsu.

**Writing – review & editing:** Ulrike Münzner, Tomoya Mori, Marcus Krantz, Edda Klipp, Tatsuya Akutsu.

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
