## [Decision Letter · Decision Letter 0]

13 Jul 2021

Dear Prof Akustu,

Thank you very much for submitting your manuscript "Identification of periodic attractors in Boolean networks using a priori information" for consideration at PLOS Computational Biology.

As with all papers reviewed by the journal, your manuscript was reviewed by members of the editorial board and by several independent reviewers. In light of the reviews (below this email), we would like to invite the resubmission of a significantly-revised version that takes into account the reviewers' comments.

All three reviewers were supportive of the manuscript and commented on it's potential impact in the field. However they raised a number of comments that are important to realize this impact. In particular, Reviewer #1 noted a number of issues in communication and clarification about the method. And Reviewer #2 and 3 raised a number of technical issues about the strengths and weaknesses of the approach that need to be addressed. Suggestions were also raised about the code. As long as the code is provided as indicated on Github these are more suggestions, as they would increase the ability of users to apply the methods.

We cannot make any decision about publication until we have seen the revised manuscript and your response to the reviewers' comments. Your revised manuscript is also likely to be sent to reviewers for further evaluation.

Sincerely,

Jeffrey J. Saucerman

Associate Editor

PLOS Computational Biology

Jason Haugh

Deputy Editor

PLOS Computational Biology

Reviewer's Responses to Questions

**Comments to the Authors:**

Reviewer #1: Dear authors,

Reading your paper on the development of efficient detection of cyclic attractors in large Boolean networks based on a priori information about the expected state of a subset of nodes convinced me that your approach is innovative and potentially quite useful, as we as a field scale up to Boolean models of increasing size and incredibly unmanageable attractor repertoire. Thus, most of my comments below are ideas on how to articulate the merit of this method more clearly. I believe improving the clarity of the paper and describing the need and applications for your method more compelling would significantly improve your manuscript.

1. Articulating your motivation:

-- I appreciated the introduction articulating WHY the size of the state space of potential initial conditions, 2^n, is a problem for large networks. However, instead of considerable effort to convince readers that a speedup is a good thing, I think it would be important to explain at the start WHY an algorithm a priori information about the desired state of some or all nodes is GENERALLY or very often useful. Some questions to consider here:

* When do we have this type of a priori information? Biological examples above and beyond 2 already built models would help

* How general is the situation in which your method could aid modelers examine their models in more depth than before?

Overall, I think the Introduction needs to convince the reader that this is NOT an obscure application, that it could be a useful tool in the repertoire of anyone building a large model who knows some things (but not everything) about the biological phenotype their model must reproduce in different conditions, and thus wishes to find any and all attractors that get close to these phenotypes.

-- In the Discussion, the sentence "In addition, a priori information might be useful to efficiently stabilize biological systems" should be explained in 1-2 sentences; just noting that this is the case without context (requiring a side-trip to the citation) is not that helpful; it leaves readers wondering: why? how? when?

2. Clarifying the logic of your pseudo-code:

-- I think that describing the heart of your algorithm in 1-2 sentences before introducing the pseudo-code would make reading this part of the paper much smoother. I would recommend clarifying that your method start testing for attractors from the "best" subspace of x0 ∈ {0, 1}^n, one in which all nodes match the a priori "known" final attractor pattern. If the search from this subspace fails, the method moves away from these "most expected patterns" by increasing the number of bits that don't match, one at a time. There are likely clearer ways to phrase this, but a short explanation goes a long way in helping people read the pseudocode.

-- Most important: please explain / clearly define what a "desired" attractor is! Since your algorithm does more than simply verify the existence of a cycle that goes through a specific single network state - what, exactly is the definition of a desired attractor that stops the search?

3. Making the mathematics more accessible:

-- This comment reflects the bias of someone who can follow your math but has to work quite hard to figure out what you meant to do first, and then check if this is indeed what your formulas do. I recommend a beefing up of the math section with more text, and targeting a broader audience to better match the readership of Plos Comp Bio. I am thinking of readers who do not routinely approach problems in a mathematical fashion, and thus they can't read your intentions off from your formulas, but who could nevertheless follow your logic if guided. To this end, I recommend adding explanations that set up what each step / formula is designed to accomplish, and THEN showing the formula itself. For example :

"First, we evaluate the partial sum F (n, p) of E (n, p) defined by ..."

* Explain why you evaluate these sums. Explain what is it that Fn counts and were does each of the terms in the sum come from. Think of it as explaining it to a student, since your audience is likely looking for a quick understanding of your logic rather than a math puzzle.

-- I recommend a similar introduction to every step in your proofs. These manipulations quickly become trivial with use, but can represent a steep initial barrier to a large swat of your potential readers.

4. Improving figure captions:

-- Figure captions should clearly explain WHAT is shown on the figure. For example, in Fig S2: what do the on/off panels correspond to? These aspects of the figure should be clearly listed rather than inferred from the main text. In contrast, the explanations of what the results on a figure mean need to be in the main text. Please recheck each figure and make sure the caption explains what you show.

5. Improving the description of prior models:

-- I have found the introduction of the example networks confusing, especially with respect to why these models are good case-studies for an algorithm with a priory knowledge. In addition, I often felt that these sections do not stand on their own, they expect the reader to be intimately familiar with the two papers the models were published in. Some things to consider in order to improve this for BOTH models:

* How did the authors of the original study find their attractors, what did they miss and why?

* Most important: what, precisely, is the prior knowledge you use, and why do you expert it to improve on their results (why should this part of the state space you single out for priority search lead to novel / more relevant cycles?)

* How / when does this scenario generalize to other models?

-- Specifically explain / clarify the following:

* Angiogenesis model: "induces typical behavior if all detected attractors correspond to a single EC behavior" -- what does this mean and is this the "typical" behavior (it would help to know what the 16 inputs are, and to make the biology of this model a bit more clear.

* Angiogenesis model: "We could identify periodic 375 attractors in the angiogenesis BN with 142 nodes using a semi-informed initial guess." Please explain your " semi-informed initial guess" before presenting the results of your algorithm.

-- Which nodes are set to 0/1 with what probably, and why?

-- When do you stop the algorithm because you have found a desired attractor?

* Cell cycle model: what is a DESIRED attractor that stops your algorithm's search?

* Cell cycle model: "then analyzed the periodic attractors (S1 Table) and found that one corresponds to the wildtype, 39 correspond to viable deletion mutants"

-- What do you mean by this? which model are you analyzing, the WT or the mutant one? If the WT, how and why would some of its attractor correspond to viable deletion mutants? If you run both, why would you report the attractors in aggregate?

* Cell cycle model: "an invalid pattern with no biological meaning" -- what does this mean, why invalid, and how is this distinguished form attractors that do have biological meaning? What is their phenotype? What does it tell you about the model, or a mutant version, that it has these non-biological attractors? (especially since this appears to be different from a "lethal" phenotype ...; are there supposed to be stable apoptotic attractors in this model, which are biologically relevant, just "dead"? )

-- The sentence: "Finally, we used 10 random states as initial guesses, and only detected point 328 attractors, regardless of switching the input Nutrients to on or off" brings up a few issues

* clarify what method you used to start from 10 random states, and find 238 point attractors. Was this your ATTapriori algorithm? In that case, what does "10 random states as initial guesses" mean, exactly?

* Why 10 initial conditions? how does that compare to other attempts used by you or the authors of the original study?

* In my experience, a synchronous Boolean model with a cyclic attractor that is supposed to represent a robust biological rhythm needs to have a pretty large attractor basin leading to that limit cycle. Ideally, the limit cycle or something quite similar to it should show up even under asynchronous update! Does this sentence indicate that the original, published biological limit cycle itself is very difficult to find without prior knowledge? If yes, this questions the validity of the original model as a robust cell cycle model; if no, please clarify why not.

-- Please explain your work with mutant genotypes and phenotypes more clearly, especially: "detected mutational genotypes with a directly interpretable corresponding phenotype based on the type of detected attractor. That is, the algorithm automatically performed an in silico mutagenesis screen, a task which had to be performed manually previously."

* You appear to claim that the algorithm finds mutant genotypes automatically. If you mean genotypes -- how? Mutant phenotypes require a change in the network structure or logic rules; in what way does your algorithm do this, how does it test the phenotype of these mutant models, and how dies it check if it matches known biology (or predicts a mutant phenotype, if it is yet untested)? How and why do you claim an automatic mutant screen?

* How do you interpret the mutant phenotype?

I hope you find these comments helpful and constructive. I think your work deserves attention form anyone wrangling a network larger than 25 nodes (so, many of us), and thus a convincing paper that is easy to read can help.

Reviewer #2: Review for “Identification of periodic attractors in Boolean networks using a priori information”.

In this manuscript the authors study the problem of finding specific attractors of Boolean networks when information about the 0/1 patterns are known with certain probability p. A formula in terms of p is derived that shows the expected number of trials to find such attractor. It is shown that if p>0.5 then the expected time to find an attractor is lower than the expected time using an exhaustive-search approach.

To the best of my knowledge this idea is new and the results are valid. I also tested the submitted codes with one of the examples and it seems to work well. However, there are some questions/concerns that need to be addressed.

About the algorithm/implementation

1. It seems that the time complexity is for finding ONE attractor only. What can be said about finding ALL attractors for which there is a priori information? There seem to be something about that in the manuscript, but it is not clear.

2. The Github site is empty. The code should be there since the manuscript states "The ATTapriori algorithm is implemented in C, and the source code is available at https://github.com/takutsu5/ .”

3. It is strange that the code internally uses

load_net("network.boolnet");

load_init_val("init_vals.txt");

instead of just reading from user input when the compiled code is used. This means that a user has to edit and compile the source code every single time to able to use it for a different example. It would be easier if the code is compiled once and then used as

./AttPrior network.boolnet init_vals.txt >attlog2

or similar. Is there a reason for the current implementation that requires modifying the source code?

About the examples.

4. [This is a major concern] It is not clear what was the advantage in using the proposed algorithm in the examples. Was it faster? Were there new insights that would not be found without the algorithm? I suppose it is the former. If so, the authors should explain (in percentage, perhaps), how much faster the algorithm was compared to an exhaustive search approach. Perhaps show that a specific attractor was found X% faster with the proposed algorithm? The examples have to clearly illustrate why this algorithm was better than an exhaustive search approach at least for one attractor.

Reviewer #3: This paper proposes to use prior knowledge to accelerate the identification of attractors in Boolean networks. It uses prior probabilities as an heuristic to reduce the expected number of states to explore before finding the first attractor. Such prior knowledge is usually not available but it could correspond to expected behaviours based on experimental observations or hints obtained from other, less complex, models. The authors discuss the use of their approach on random networks and on two large models, along with some new biological insight. Overall the topic of the identification of attractors is important and the proposed method makes sense in some cases. I did not fully check and the analysis of the expected complexity, but the general principle seems valid, but be based on some assumptions which are not stated very clearly. However I find some of the claims presented here a bit excessive.

The current state of the art approaches rely on constraint solving. In practice constraint solvers explore many branches and it is indeed true that for some very peculiar models they would have to explore the whole state space. However actual models enable to cut large parts of the search space: for example if a component A is required for the activation of a component B, then branches with A fixed at 0 and B fixed at 1 are not explored, not just because they are less likely to contain a solution, but because it can structurally not contain one. The expected complexity is in practice much better than O(n^2). Several methods exist for stable states (using prolog, SAT solvers, explicit decision diagrams or ASP), as well as the approximations of complex attractors through stable motifs (also called trap spaces or symbolic steady states, which are missing from the paper).

Note that using these approaches, the set of constraints could be extended to restrict the search to attractors matching some expected patterns, and the absence of matching attractor could then be formally verified without exploring all states. However this involves some knowledge about the inner working of these methods.

In the introduction, the authors argue that no method have a worst case complexity better than O(n^2) (i.e. enumerating all states). This is true.

Then they claim to break this limit using prior knowledge. Here they compare their theoretical EXPECTED complexity to the theoretical WORST case of other approaches. Then they only search for a SOME attractor while other approaches target the formal identification of ALL of them. As the method relies on enumeration plus simulation, it's theoretical worst case complexity is still O(n^2).

Note that in deterministic models, finding at least one attractor assuming that it has a short length has always been a fairly easy problem: pick some initial state (random or not) and perform a simulation with a rotating memory until recovering the last memorized state (with some ways to avoid looping forever if the memory is too short). This work seems to mainly propose the use of prior knowledge as heuristic for the initial sampling of initial states. It can be useful for models too large for a more formal analysis, but it is limited to deterministic models (finding attractors through simulation in the (more realistic) non-deterministic case is much harder) and gives less informative results.

The description of the sampling method could be improved, especially as the expected complexity is evaluated on the number of trials. The text suggests that this corresponds to the number of tested initial state but this does not make sense. Is it the length of the trajectory leading to the first state which is part of an attractor?

The described algorithm has no specified memory size and limit on the length of explored trajectories, meaning that it could explore the whole state space on the first initial state and still end up finding an attractor thus be counted as success. If the expectation is that the selected initial state is itself part of the attractor, then the memory size and the cost of comparison are small, but it needs a limit on the length to decide that it failed.

Furthermore, the examples in the explanation use probable states of 0, thus the optimized sampling order is an order which could well be used by default in absence of prior information.

Last, the paper mentions a check that the found attractor is "a desired one" without really saying what this means. Is it a comparison to the probabilities used as input? What would be the error margin for accepting it? Is this score computed during the simulation and used to decide when to interrupt it?

The paper contains the following statement: "most practical Boolean functions can be evaluated in polynomial time and the lengths of trajectories in most practical BNs are considered to be not very long". If this is true then a simulation from any initial state would lead to an attractor in a short time. THis is often true, but not a formal guarantee. I suspect that the advantage of the proposed approach is to shorten the trajectory leading to an attractor by selecting a good initial state.

I feel that a more extensive comparison of the total number of explored states until the discovery of the first attractor with and without prior knowledge (and using bad prior knowledge) is needed to evaluate the effect of prior knowledge (unless this is what figure 2 is supposed to show?).

During the analysis on random networks, the use of "perfect" prior knowledge based on a state identified to be part of the attractor is a very strong assumption. Some noise should be added to it so that the most probable state would not be exactly part of the attractor. This may be related to this sentence that I did not understand: "The 0/1 values of each node of the detected attractors were changed into 1/0 with the reverse given a priori probability in order to generate an initial state of the random N-K networks".

The authors illustrate their approach on two models which they claim are untractable with existing tools. I did some tests using the original model of endothelial cells (with 142 components, it is much smaller than the cell cycle model). Using bioLQM it was possible to identify all 1'097'952 attractors of this model within 4 seconds on a laptop (using decision diagrams, note that printing the result takes significantly longer). Within 20s, I could identify the first 100'000 trap spaces of the same model using PyBoolNet (using an ASP solver). With it I could also pick a random initial state, identify the 45 components which are structurally fixed in this state and use this to find the 2 potentially reachable trap spaces (both fixed points) within half a second. I also tested this model with BNS (using a SAT solver to identify synchronous attractors) which provided hundreds of results within one second.

The author used a modified version where all components with fixed values have been turned into input components. This strongly increases the total number of attractors but probably does not affect the difficulty of finding at least one deterministic attractor with AttPrior. With this variant, the analysis of all fixed points becomes hard, but the analytical identification of some attractor, or their full identification after setting the initial state remain easy.

Considering these results, I would say that this model remains perfectly tractable using existing tools. Despite the large number of attractors, finding all potential attractors for any specific initial state is fairly easy even in the non-deterministic case.

The second example model is much much harder to work with: loading it into bioLQM or PyBoolNet is a challenge as these tools use heavy representations for the Boolean rules. Using some personal code with lighter data structures, I could find quickly some attractors by sampling deterministic simulations without having to use prior knowledge, but it involved fairly long traces. Turning the full model into a "good" set of constraints was much much slower than finding some solutions for the set afterwards. In this case, I would indeed say that formal analysis becomes problematic and that the proposed approach is useful. I suspect that rewriting the model to replace complex rules with additional components would facilitate formal analysis, but this is not an easy task either. From my (biased) perspective, this highlights the need for improving the translation of Boolean rules into sets of constraints. The proposed approach should be used for very large models until this happens (if ever).

While testing the provided code on these two models I ran into a few problems:

* the github repository was empty until very recently, and this version does not work with the first model

* the code provided in the supplementary archive did work but needs to be compiled to change the input files (which is solved in the github version).

* while it uses the bnet format (good point for avoiding the use of yet another new format), it does not properly parse it: components with fixed values, header line and comments should be handled to be able to use existing models

The code would be useful after improving the support for the bnet format and including some examples in the github repository. A git tag to find the content of the supplementary from the repository would be nice.

**Have the authors made all data and (if applicable) computational code underlying the findings in their manuscript fully available?**

Reviewer #1: Yes

Reviewer #2: **No: **No quite. Authors state code is available in Github, but it is not. That will confuse readers.

Reviewer #3: Yes

PLOS authors have the option to publish the peer review history of their article (what does this mean?). If published, this will include your full peer review and any attached files.

Reviewer #1: **Yes: **Erzsébet Ravasz Regan

Reviewer #2: No

Reviewer #3: No
---

## [Decision Letter · Decision Letter 1]

29 Nov 2021

Dear Prof Akustu,

We are pleased to inform you that your manuscript 'Identification of periodic attractors in Boolean networks using a priori information' has been provisionally accepted for publication in PLOS Computational Biology.

Before your manuscript can be formally accepted you will need to complete some formatting changes, which you will receive in a follow up email. A member of our team will be in touch with a set of requests. Please also consider the comments of Reviewer #3.

Best regards,

Jeffrey J. Saucerman

Associate Editor

PLOS Computational Biology

Jason Haugh

Deputy Editor

PLOS Computational Biology

Reviewer's Responses to Questions

**Comments to the Authors:**

Reviewer #2: Authors addressed the concerns.

Reviewer #3: This revised version is much improved, the authors now better compare their approach to the state of the art and provide a well-balanced description of its scope.

In the introduction (line 43) the authors list stable motifs, symbolic steady states and trap spaces as if they were alternative approaches, but they are only different names for the very same thing.

The authors have added a modified version of their software to the github repository but said that the supplementary archive remains unchanged as it is specialized. I think that the confusion induced by the duplication in the repository is a very high price to pay for a non-existent problem. The improved boolnet parsing support should not be a problem for the specialized use case (if the header line is detected instead of being immediately ignored) and the modified file also contains some actual logic changes (lines 928 and 1219) which may be useful in the archive as well. The duplication of the readme file should also be avoided (the github interface shows the old readme file).

**Have the authors made all data and (if applicable) computational code underlying the findings in their manuscript fully available?**

Reviewer #2: Yes

Reviewer #3: Yes

PLOS authors have the option to publish the peer review history of their article (what does this mean?). If published, this will include your full peer review and any attached files.

Reviewer #2: No

Reviewer #3: No

---

## [Editor Report · Acceptance letter]

10 Jan 2022

PCOMPBIOL-D-21-01094R1 

Identification of periodic attractors in Boolean networks using *a priori* information

Dear Dr Akutsu,

I am pleased to inform you that your manuscript has been formally accepted for publication in PLOS Computational Biology. Your manuscript is now with our production department and you will be notified of the publication date in due course.

With kind regards,

Olena Szabo
